# An initial report of circa 241,000- to 335,000-year-old rock engravings and their relation to *Homo naledi* in the Rising Star cave system, South Africa

Lee R Berger[1,2,3]*[†], John Hawks[2,4][†], Agustín Fuentes[2,5][†], Dirk Van Rooyen[2,6], Mathabela Tsikoane[2,6], Maropeng Mpete[2,6], Samuel Nkwe[2,6], Keneiloe Molopyane[2,7][†]

[1]The National Geographic Society, Washington DC, United States; [2]Centre for the Exploration of the Deep Human Journey, School of Anatomical Sciences, University of the Witwatersrand, Johannesburg, South Africa; [3]The Carnegie Institution for Science, Washington DC, United States; [4]Department of Anthropology, University of Wisconsin, Madison, United States; [5]Department of Anthropology, Princeton University, Princeton, United States; [6]The National Geographic Society Rising Star Project, Cradle of Humankind, UNESCO World Heritage Site, Johannesburg, South Africa; [7]Geography, Archaeology and Environmental Studies, University of the Witwatersrand, Johannesburg, South Africa

*For correspondence: lrberger@ngs.org

[†]These authors contributed equally to this work

## eLife Assessment

This article presents **important** information about potential *Homo naledi*-associated markings discovered on the walls of the Hill Antechamber of the Rising Star Cave system, South Africa. If confirmed, the antiquity, intentionality, and authorship of the reported markings will have profound archaeological implications, as such behaviors are otherwise widely considered to be unique to our species, *Homo sapiens*. This report concerns preliminary findings, and as it stands the study is **incomplete**, with further work needed in the future to support the claims about the anthropogenic nature, age, and author of the engravings.

**Abstract** The production of painted, etched, or engraved designs on cave walls or other surfaces is recognized as a major cognitive step in human evolution. Such intentional designs, which are widely interpreted as signifying, recording, and transmitting information in a durable manner, were once considered exclusive to Late Pleistocene *Homo sapiens*. Here we present observations of what appear to be engraved abstract patterns and shapes within the Dinaledi Subsystem of the Rising Star cave system in South Africa, incised into the dolomitic limestone walls of the cave. The markings described here are found on a pillar in the Hill Antechamber that extends into the natural fissure corridor that links the two chambers, and we associate them with *Homo naledi*. They include deeply impressed lines, cross-hatchings, percussion marks, and other geometric shapes on flat wall surfaces and in and around existing cracks and grooves in the dolomitic limestone walls, found in one specific location of the Dinaledi Subsystem. Remains of multiple *H. naledi* are found in this part of the cave system, and evidence of mortuary behavior appears in both the Dinaledi Chamber and adjacent Hill Antechamber dated to between 241 and 335 ka (Dirks et al., 2017; Robbins et al., 2021; Berger et al., 2025).

## Introduction

The Rising Star cave system, South Africa, is located within a small promontory situated to the south and east of the course of the Blaaubankspruit stream. The cave system is situated within the dolomitic limestone of the Malmani Subgroup, a Precambrian marine rock bedded with chert bands and containing abundant stromatolite fossils (*Dirks et al., 2015*). The system includes more than 3 km of mapped passages comprising multiple levels within a west-dipping dolomite horizon. Abundant remains of *Homo naledi* (*Berger et al., 2015*) occur within several localities in the system, including the Dinaledi Subsystem, which lies at a depth of ~30 m below the present surface and ~120 m through the cave system from the nearest present entrance (*Hawks et al., 2017*; *Elliott et al., 2021*). Remains of *H. naledi* have been recovered and excavated from the Dinaledi Chamber, Hill Antechamber, and adjacent spaces and fissures (*Berger et al., 2015*; *Berger et al., 2025*; *Brophy et al., 2024*). These spaces are challenging to enter and navigate, and exploration of them is ongoing (*Berger et al., 2025*; *Elliott et al., 2021*).

On July 28, 2022, during a survey of the Dinaledi Subsystem, we identified what appear to be engraved markings on the southern and northern faces of a natural pillar that forms the entrance and exit of a narrow passage connecting the Hill Antechamber with the Dinaledi Chamber (*Figure 1*). The marks include linear features between ~5 and~15 cm in length. Many intersect to form geometric patterns such as squares, triangles, crosses, and X's, while some are isolated lines. These markings are located on three dolomitic panels on otherwise smooth areas, as well as in and around existing cracks and grooves in the rock. They occur in three areas which we have labeled *A, B,* and *C*. These markings are in a location where they can be viewed during access into and egress from the Dinaledi Chamber when entering the system from the Hill Antechamber. The Hill Antechamber is the likely point of access by *Homo naledi* to the entire subsystem, and the passage is the natural linkage between the two main chambers of the subsystem (see *Figure 1*, also *Elliott et al., 2021*).

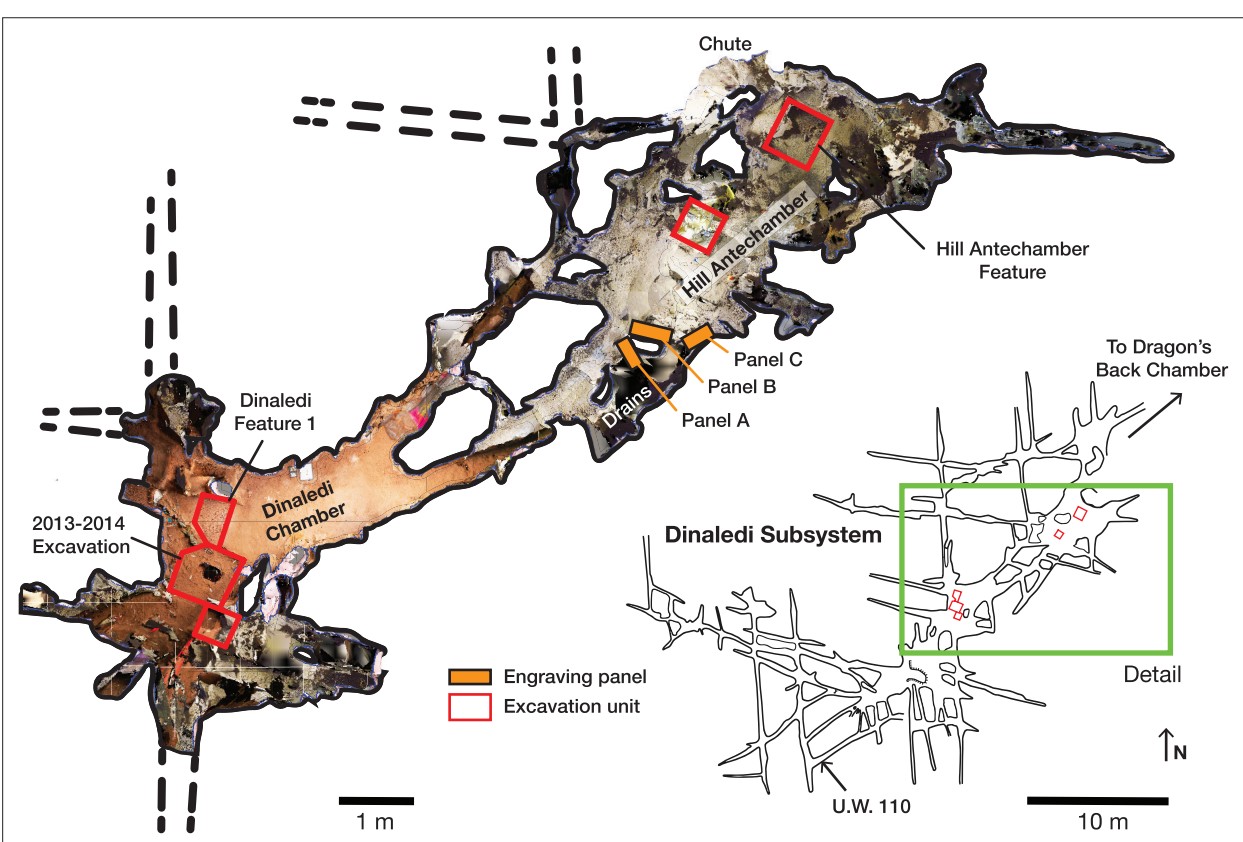

**Figure 1.** A photogrammetric map of the Dinaledi Subsystem of the Rising Star cave system, South Africa. Orange bars mark the positions of the Engravings panels on the walls of the cave. Red boxes outline areas where excavations have been undertaken in the floor of the chambers. The green box outlines the area enlarged and colorized.

In this article, we describe our observations of Panel A within the passage linking the two main chambers. We provide images of Panels B and C within the Hill Antechamber and offer suggestions of their contextual relationship with Panel A, while recognizing that identifying and examining any and all potential engraved lines within these panels requires further study. We also provide additional contextual data supporting the attribution of these markings to *H. naledi* and discuss the time sequence represented by engraved marks on Panel A. We end by discussing potential implications of our findings for *H. naledi* culture and cognition. The description and initial analyses provided here are intended to document the discovery and provide spatial and contextual information to the scientific and heritage community prior to any further analyses that may require invasive sampling as such may require destructive methods or interacting with the marks directly, and therefore have associated ethical implications that need to be carefully considered before being undertaken (e.g., *Bednarik, 2008*; *Arthur et al., 2021*; *Zerboni et al., 2022*).

## Results
### Panel A
What we term Panel A is found on the southern face of the natural pillar that forms the southern edge of the entry from the Hill Antechamber into the southern of two passages leading to the Dinaledi Chamber (*Figures 1 and 2*). The panel is notable as an area of discolored rock that appears to have been smoothed by percussive blows by a hard object, as is evidenced by micro and macro pitting of the surface not found on the adjacent natural rock surfaces (*Figures 3 and 4*) and by the possible application of sand and grit both before and after markings were made (*Figures 5 and 6*) and (*Figures 7–9*) .

The most visible markings are engraved lines, some of which when viewed together are crosshatched and give the impression of a rough hashtag figure (*Figures 10 and 11*). The lines appear to have been made by repeatedly and carefully passing a pointed or sharp incising tool into existing or new grooves. In addition, there are markings that are best described as scratches that fall outside of identifiable lines and their intersections. Several of the grooves overlap geological features native to the rock, including fossil stromatolites (*Figures 12 and 13*). In many instances, we suggest it is possible to identify which lines were made first by examining the point where they cross another line (e.g., *Figures 14 and 15*).

We identified at least 46 engraved or altered marks we consider artificial on Panel A (see *Figure 16*). In this number, we include features that appear to be natural lines that have been altered by etching. The most prominent markings on Panel A are a series of intersecting lines (*Figures 10–12*). There appears to be a temporal span involved in the creation of the engraved lines as some seem more recently engraved and show clean etching, while others have been obscured either by slight weathering or by the application of sediment. The most easily identifiable engravings, based on their clarity, are lines L2, L6, L9, L11, L16, L17, L27, L30, and L31 (*Figure 16*). While the existing lines may have been created in older etchings or been created over multiple interactions, the final etchings of the lines based on which lines overlap we interpret as follows: horizontal lines 11 and 25 were created after vertical line 2. Vertical line 6 was created after L11. Vertical line 18 was created after horizontal line 17. Line 30 was created after horizontal line 25, but before horizontal line 31.

## Evidence of hominin manufacture of engravings on Panel A
Dolomite is known for a pattern of natural weathering that results in patterns of recessed linear features on its surface often called 'elephant skin' patterning. Natural fissures and erosional features in weathered dolomite surfaces due to chemical weathering processes (*Burger, 1989*) are in our observations characteristically deeper than several millimeters, and they follow natural fracture planes within the rock. Natural erosional features in dolomite may have variable cross-sections, ranging from beveled to U-shaped to rectangular in cross section with rounded or flat bottoms (*Jeelani et al., 2018*) and are typically a few millimeters to many centimeters in depth, but do not have multiple parallel striations visible within them which would be characteristic of retouching or engraving. As opposed to natural surface weathering, artificial lines are limited in depth and extent due to the natural hardness of dolomite. This hardness (see *Tabor, 1954*) means that any substantial artificial marking requires multiple parallel incisions with a hard tool. Where artificial engraved markings intersect, they often appear to exhibit an ordering in which one was completed before the other (e.g., see *Figures 6 and 13–15*);

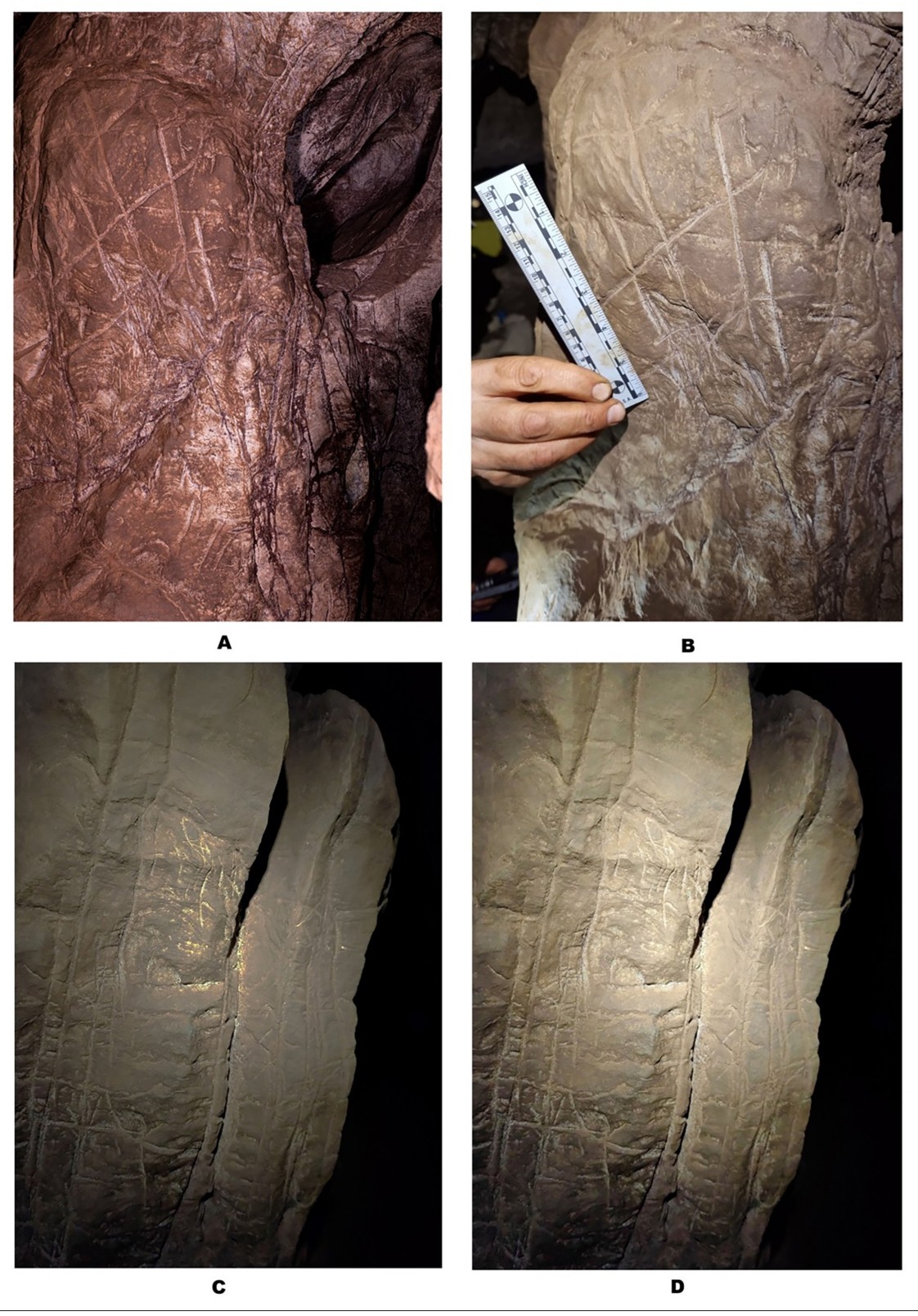

**Figure 2.** Engraving Panel A (**A, B**) and Engraving Panel B (**C, D**). (**A**) was taken with a polarizing filter as described in the 'Methods'. (**B**) was taken using only LED lights and approximates natural coloration. (**C**) shows the results of increasing contrast while lowering light on Panel B while (**D**) illustrates Panel B under LED lighting.

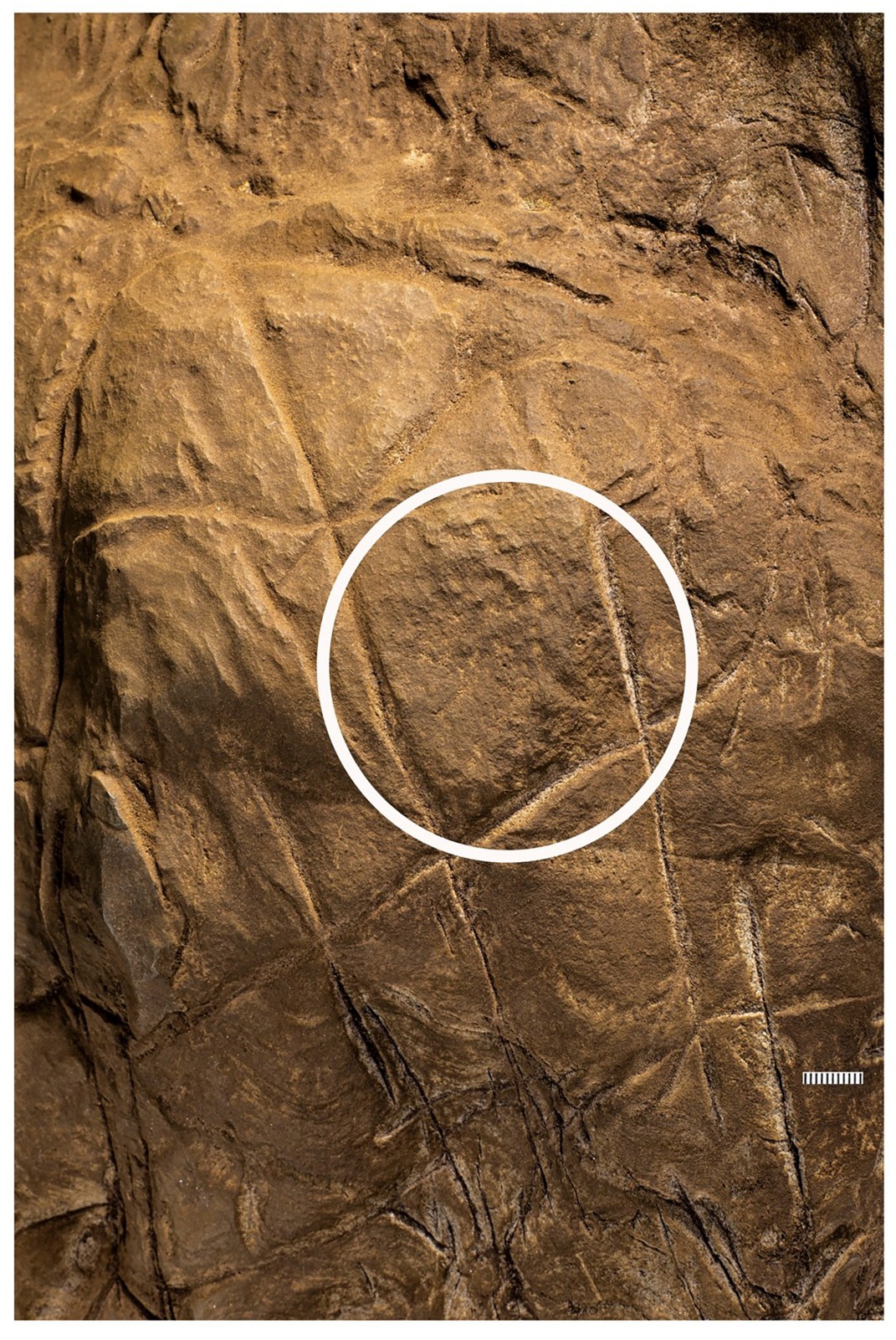

**Figure 3.** Cross-hatched etchings in Panel B. The white circle outlines areas of the engraving that may indicate hammer blows or pounding marks as evidenced by pitting not seen on other surfaces.

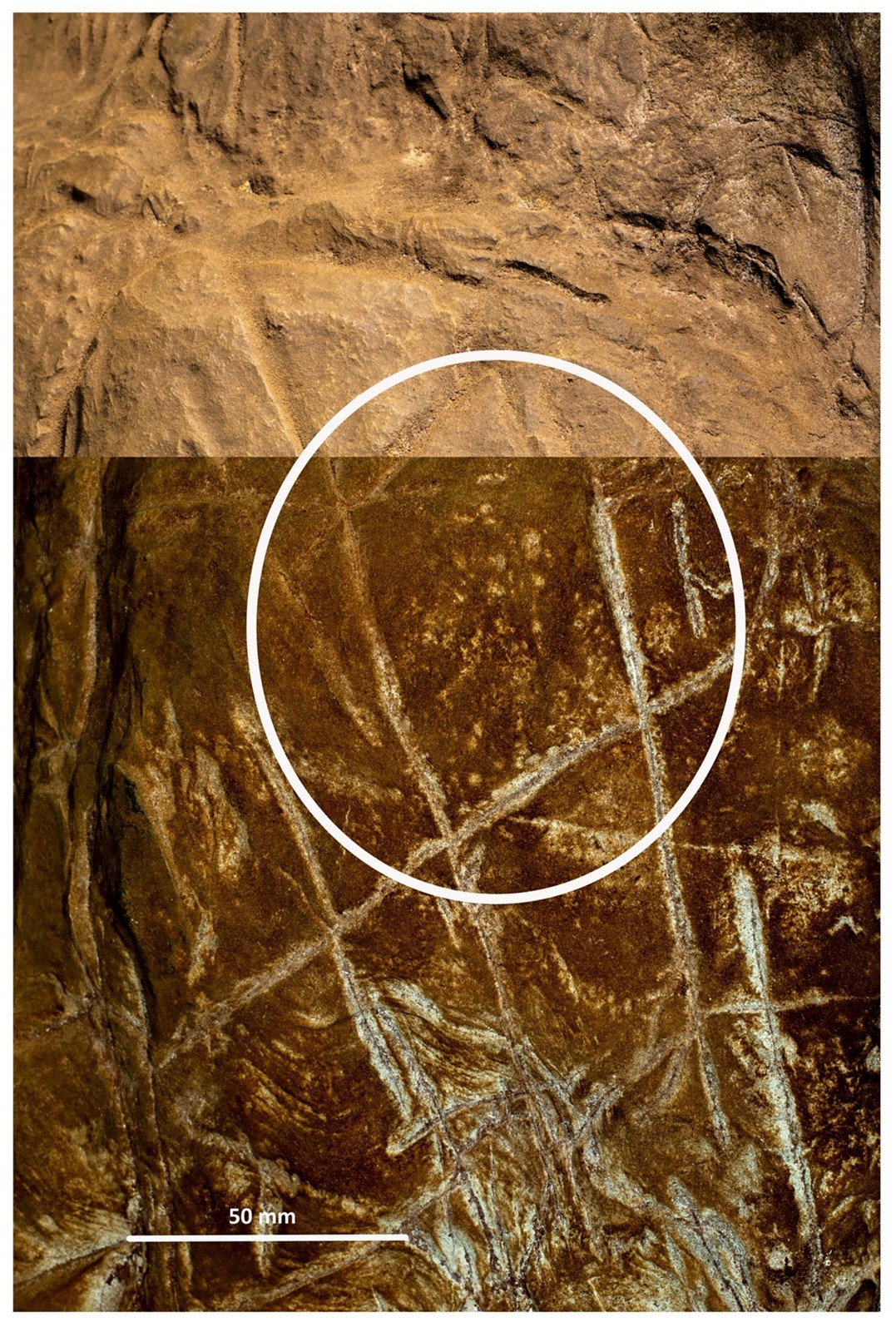

**Figure 4.** Cross-hatched etching comparing polarized images (bottom) with non-polarized imaging of the same area highlighting pitting marks that appear to be non-natural in origin.

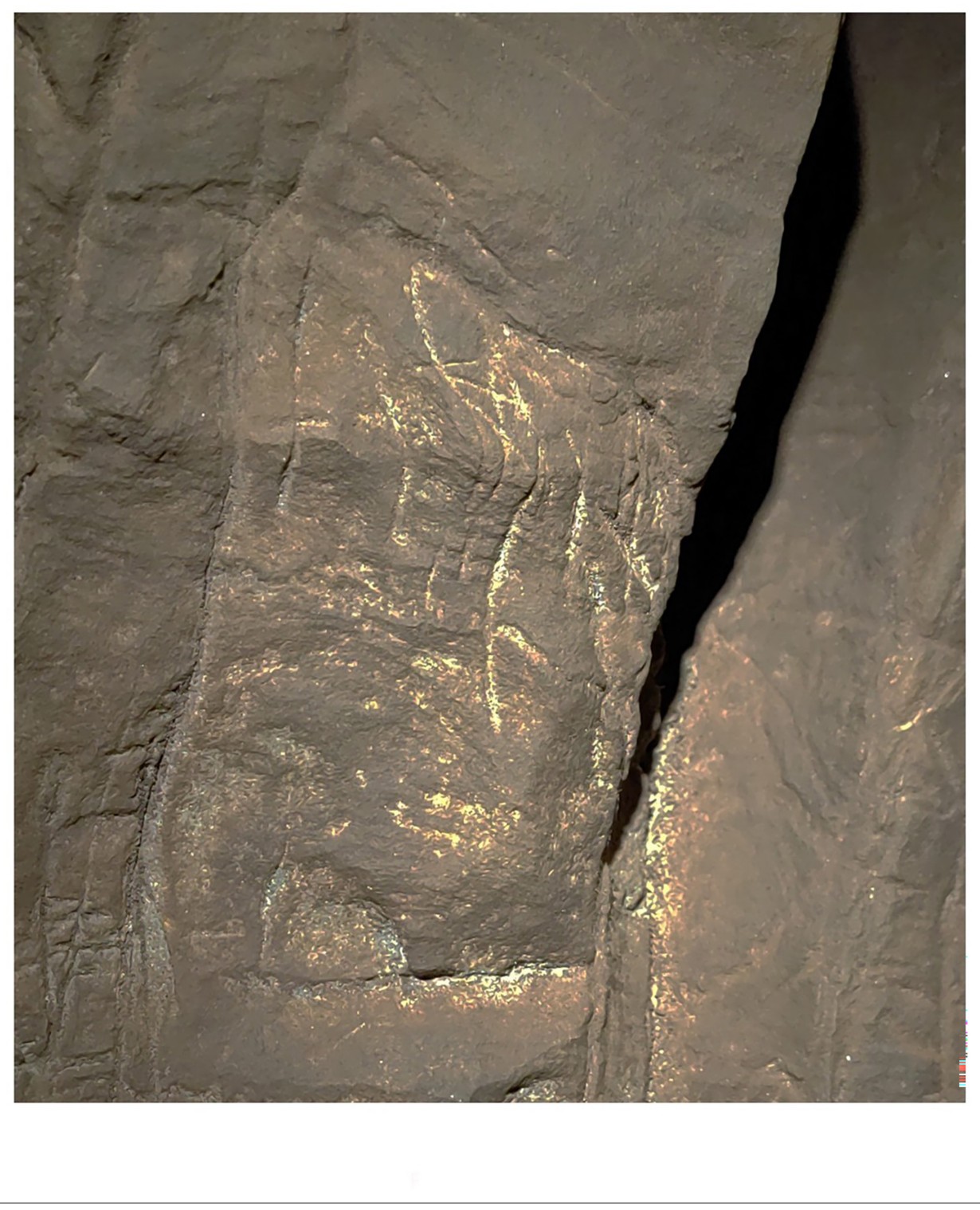

**Figure 5.** Closeup of non-geometric figure at the top of Panel B. Note the cross-like etching to the left of the figure as well as the X etched to the right. The non-geometric figure uses in part a natural fracture as an extension of the line beneath it before an inverted Y is etched at the terminus of this line. The material causing discoloration of the surface has not been analyzed.

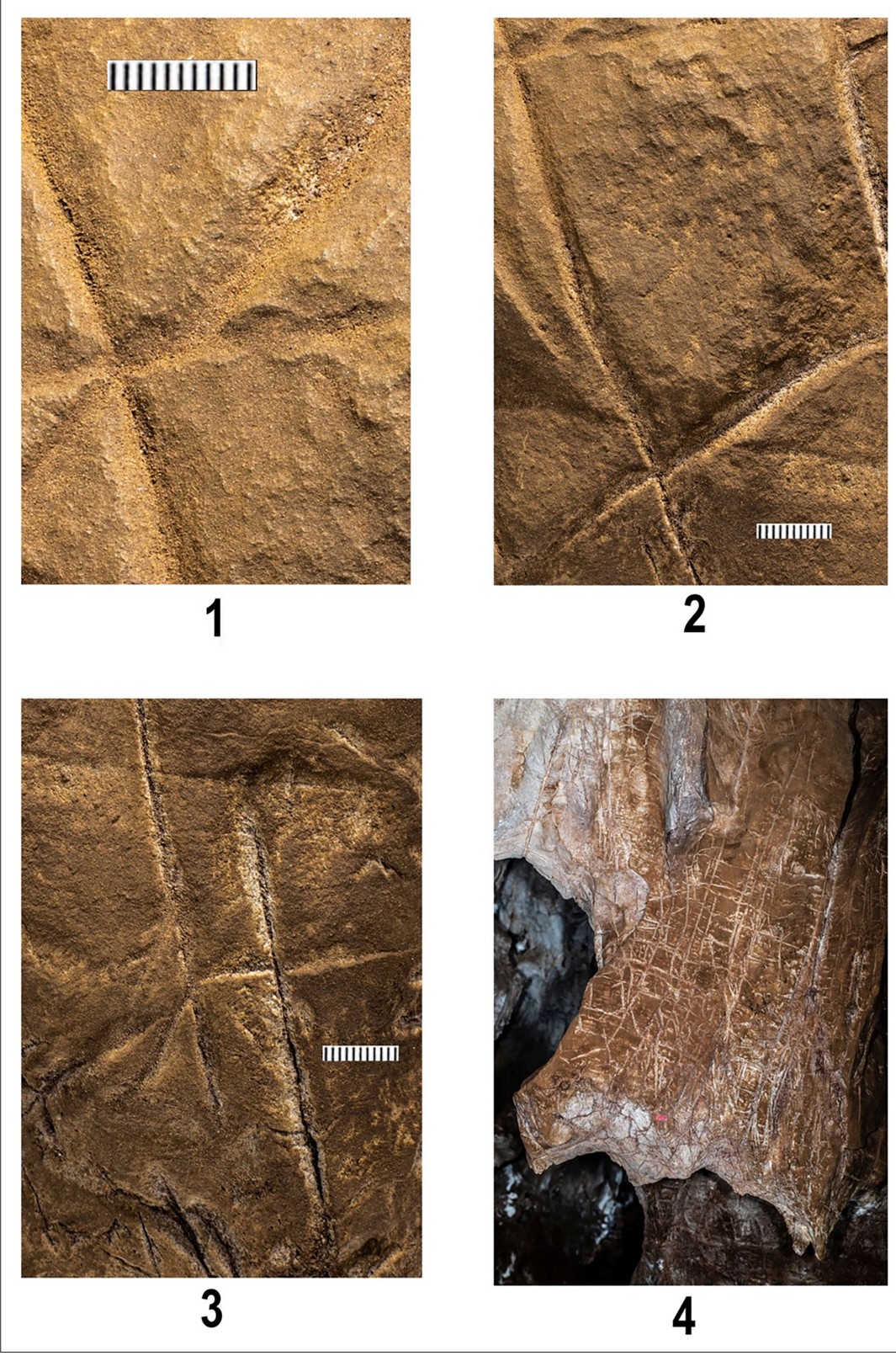

**Figure 6.** Evidence for sediment covering lines on Panel A. Image 1 shows lines 2, 3, and 4 of Panel A (see *Figure 1* for map). Note small sediment granules in the base of lines to the left of the image, while the line rising to the upper right shows penetration to the native underlying rock by the carving action. Image 2 illustrates a position slightly lower on the cross-hatch marks on Panel A imaging lines 3, 4, 6, 11, and 13. Note the difference

*Figure 6 continued on next page*

*Figure 6 continued*

between etching marks on the lower part and right of the image of this section of the engraving showing the difference between highly etched lines versus one presumably covered by a light layer of sediment post their creation. Note also likely pitting or presumed hammerstone marks in the central part of image 2 between the carved lines. Image 3 illustrates lines 6, 17, and 18. Note the sharply carved lines on the right and the lines on the left that appear to be obscured by a light application of coarse sediment. Image 4 illustrates a wider shot of Panel B showing the discoloration of the area containing the engravings compared to the native rock with no sediment visible in the upper left of the image.

this appearance of sequencing is not typical of natural weathering. In previous work, researchers have noted the limited depth of artificial lines, their composition from multiple parallel striations, and their association into a clear arrangement or pattern as evidence of hominin manufacture (***Rodríguez-Vidal et al., 2014***).

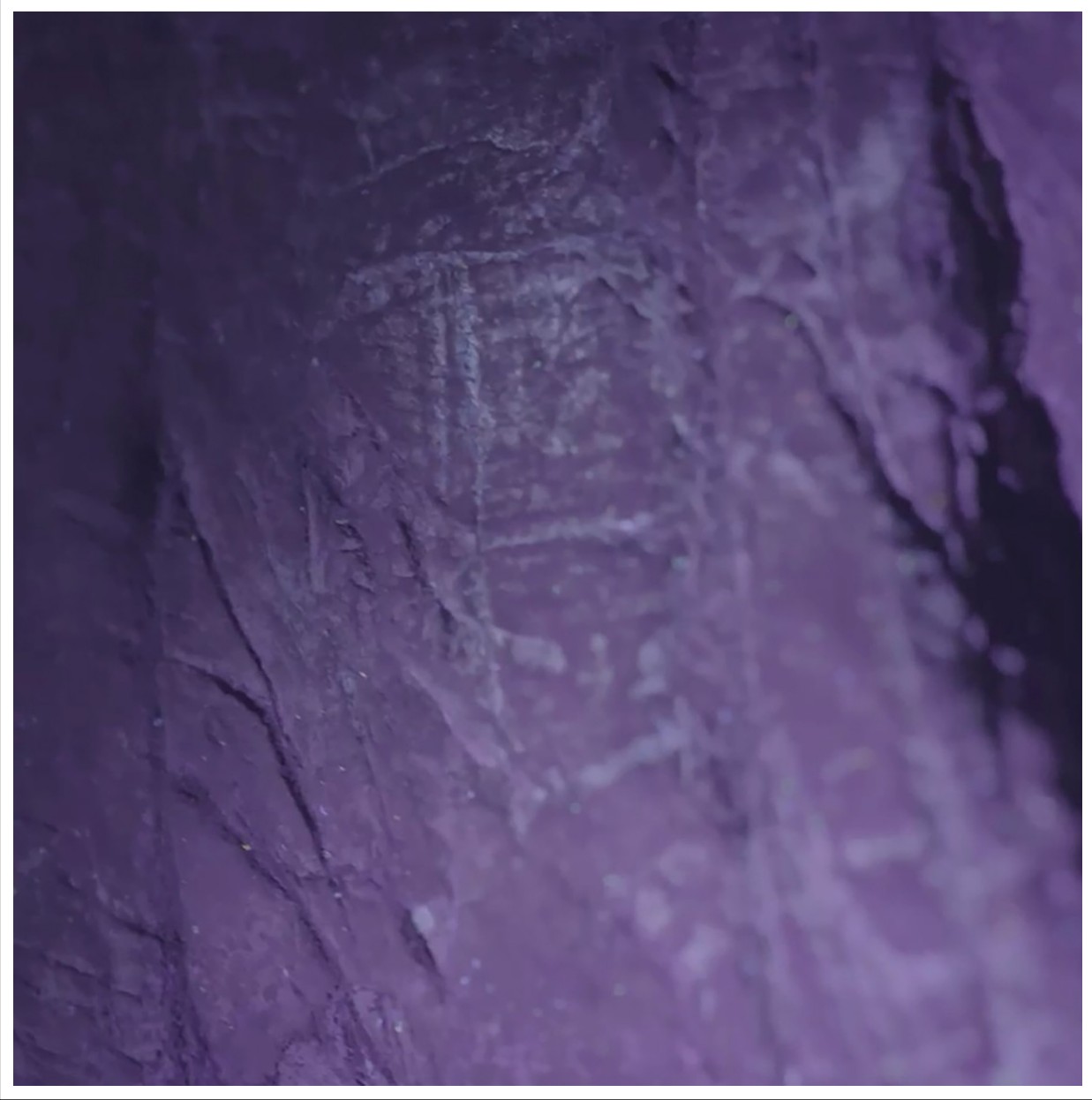

**Figure 7.** Block-like etchings on Panel B seen under ultraviolet light. Note the slight white appearance of the etched lines indicating the presence of a reflective or slightly fluorescing material in the engraved lines similar to the fluorescence of pure $CaCO_3$.

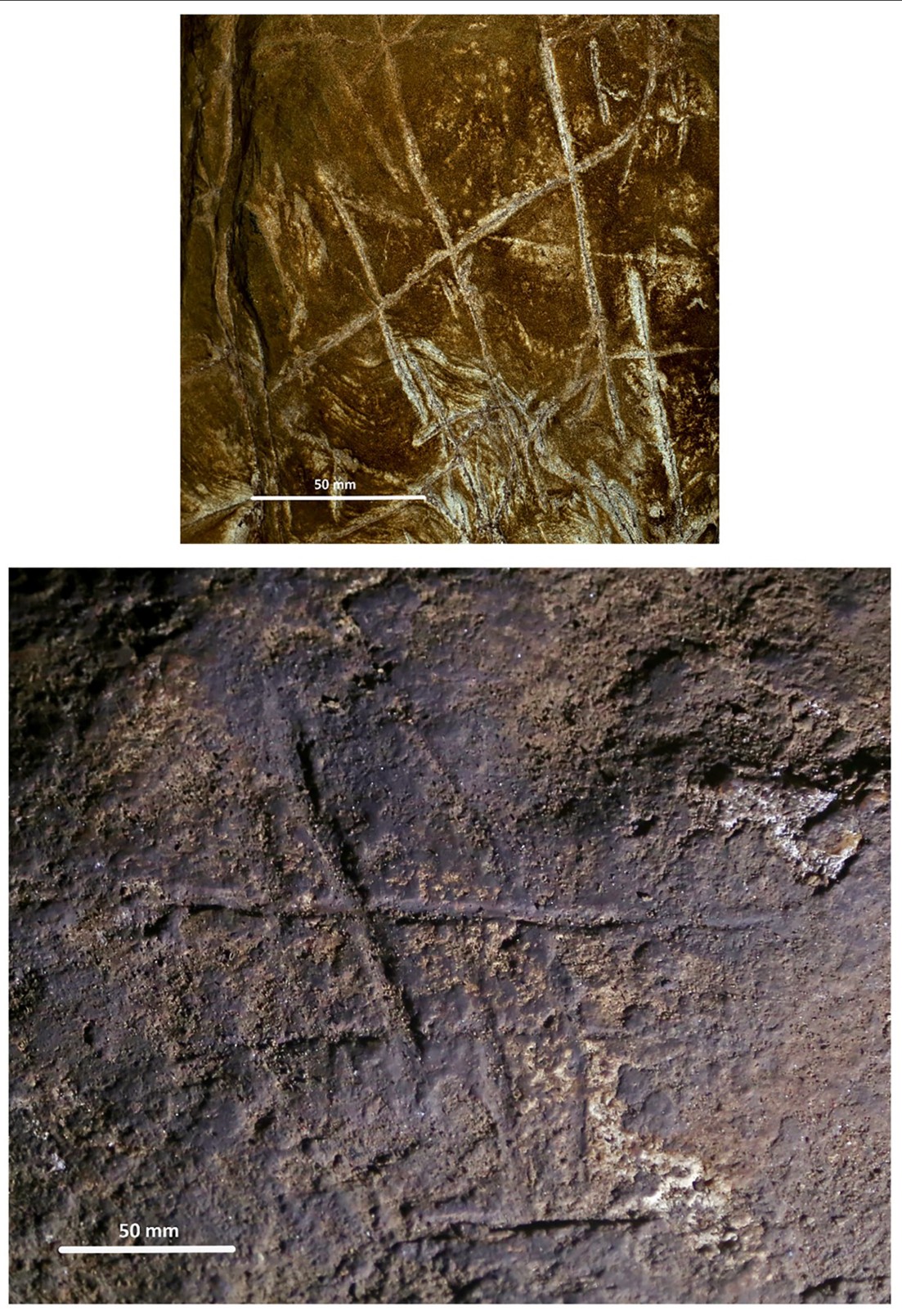

**Figure 8.** The Dinaledi Subsystem etched cross-hatch found on Panel A between the Hill Antechamber and Dinaledi Chamber (top) compared to the cross-hatch engravings found on the cave floor of Gorham's Cave, Gibraltar, and attributed to manufacture by a Neanderthal circa 60k years ago.

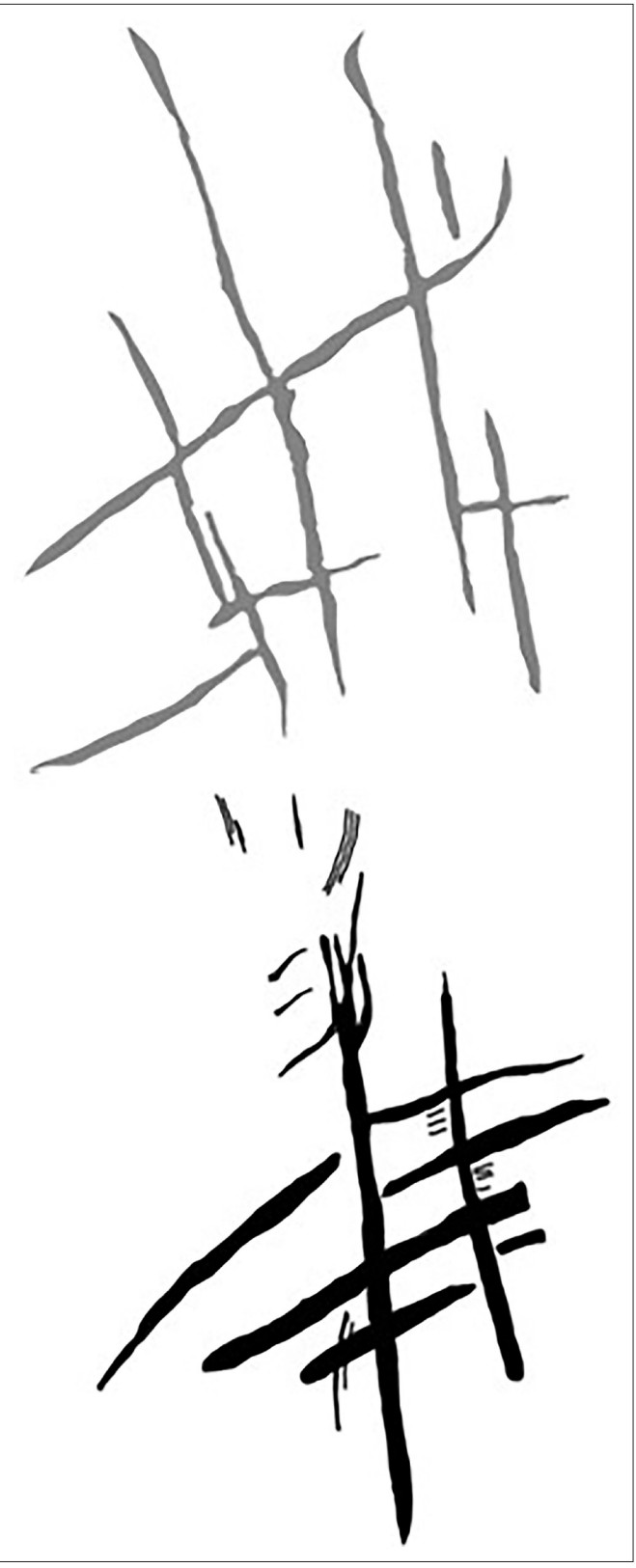

**Figure 9.** Line tracing of the Panel A Dinaledi Subsystem engraving (top in gray) compared to a line tracing of the Gorham's Cave engraving (bottom in black). Tracings not to scale.

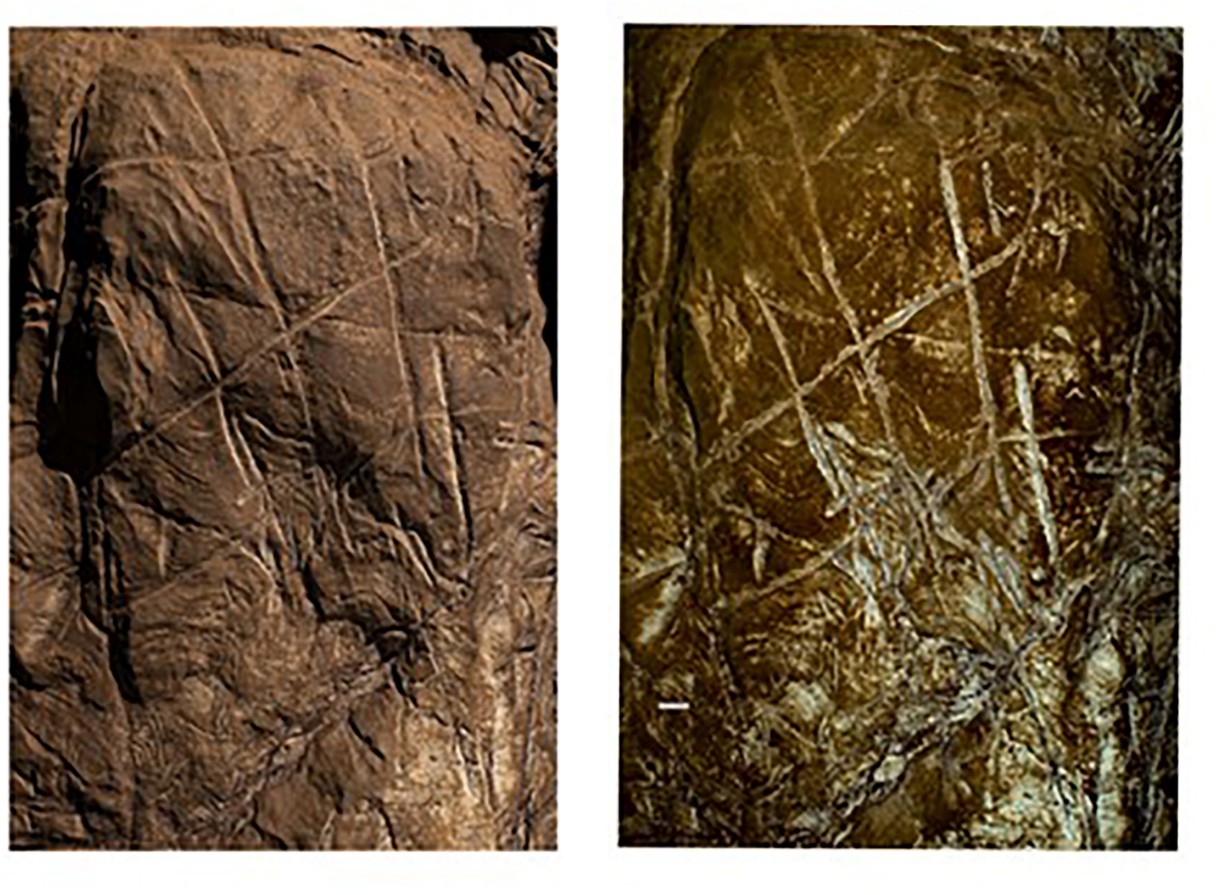

**Figure 10.** Non-polarized (left) and polarized (right) image of the cross-hatched engraving on Panel A, Dinaledi Subsystem. Scale in millimeters.

The engraved lines in Panel A diverge in their patterning from the natural 'elephant skin' weathering of the surrounding dolomite walls, which can be seen adjacent to the panel within 20 cm of the nearest artificial marks (Figure 18). Our assessment of the features produced by natural weathering in this area is that they are deeper than 10 mm, in particular deep relative to the feature width, maintain a consistency of size and depth across substantially undulating or rugged surfaces, and expand from natural cracks and fissures. In contrast, even the widest of the engraved lines that constitute Panel A have a relatively shallow depth. High-resolution macro-photography shows micro-striations constituting several of these engraved lines, in which roughly parallel incisions sometimes overlap with each other (see *Figures 6, 13–15 and 17*). Many of the lines also fall out of the direction of natural fracture features in the dolomite, although it should be recognized that there are multiple places on this panel where natural lines and features of the rock may have been enhanced or extended by artificial engraving. *Figures 13–15* show examples of ordering where engraved lines intersect, one having been completed clearly before the other. Some of the elements we describe as engraved lines appear to be artificial extensions or alterations of lines that may originally have resulted from natural weathering (*Figure 18*).

In addition to the engraving depth, composition, and ordering, two additional aspects of Panel A markings distinguish them from natural weathering. The dolomitic bedrock of the Malmani Formation includes fossil stromatolites, which manifest as curving linear banded striations visible in the rock. Panel A includes these layered stromatolitic bands, and all engraved lines that pass below the bottom of line 14 cross over this fossil feature (see *Figures 13 and 15*). Where engraved lines cross over this feature, they retain direction, and in some cases, the multiple striations slightly diverge, suggesting that maintaining a linear engraving over this irregular surface may have been challenging. Second, the markings are, in places, covered wholly or partially in sediment or some other substance. This coating does not occur in other areas of the chambers where there are no such engravings. This material (we

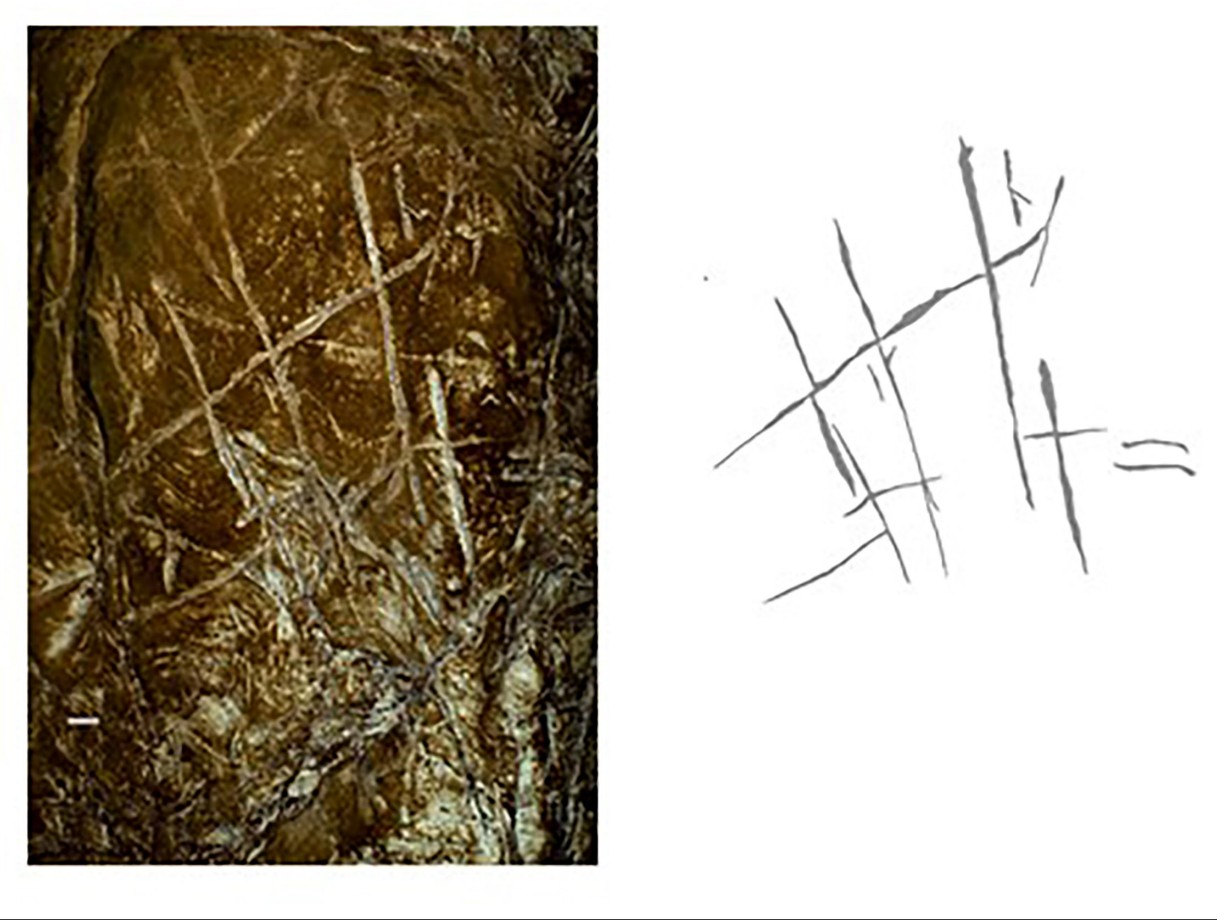

**Figure 11.** Polarized image of the Panel A engraving (left) with the most visible lines (interpreted as the most recent etchings) traced (right). Scale in millimeters.

cannot determine if it is sediment or pigment) may have been used either to create visual contrast on the gray dolomite, to abrade the surface as a form of polish, to enhance or obscure some aspects of the engraved lines, or all of these. This material is present on the surface as a thin layer and is evident within some of the grooves of the lines, indicating its application after some of the marks were made. The appearance of time-ordering between engraved lines and the surface treatment may imply an origin of the engravings in multiple episodes (*Figure 6*). Thus, it does not appear this covering, likely sediment, can be explained by geological or other non-organic processes.

The means of manufacture of engraved lines in this context would have required an implement of equal or greater hardness compared to the native dolomitic limestone. Dolomite rocks of appropriate size and morphology to mark the cave walls have been recovered from surface contexts within the Dinaledi Subsystem, as have many chert fragments. At present, only one possible lithic artifact has been described in direct association with *H. naledi* remains (*Berger et al., 2025*). This 'tool-shaped rock' does resemble tools from other contexts of more recent age in southern Africa, such as a silcrete artifact with linear engraved features on it that was recovered from Blombos Cave (*Henshilwood et al., 2018*; *Figure 19*).

## Panels B and C

Panels B and C are located on the northern wall within 2 m of the Hill Antechamber burial feature described in *Berger et al., 2025*. Panel B is situated lower and to the right (west) of Panel C. Both panels appear to have been prepared in a similar way to Panel A, with possible use of cave sediment applied to the surface, giving the surfaces of these panels an obvious textural difference to adjacent walls of the chambers (*Figures 2B, C and 5*). A number of etchings and engravings can be seen, some

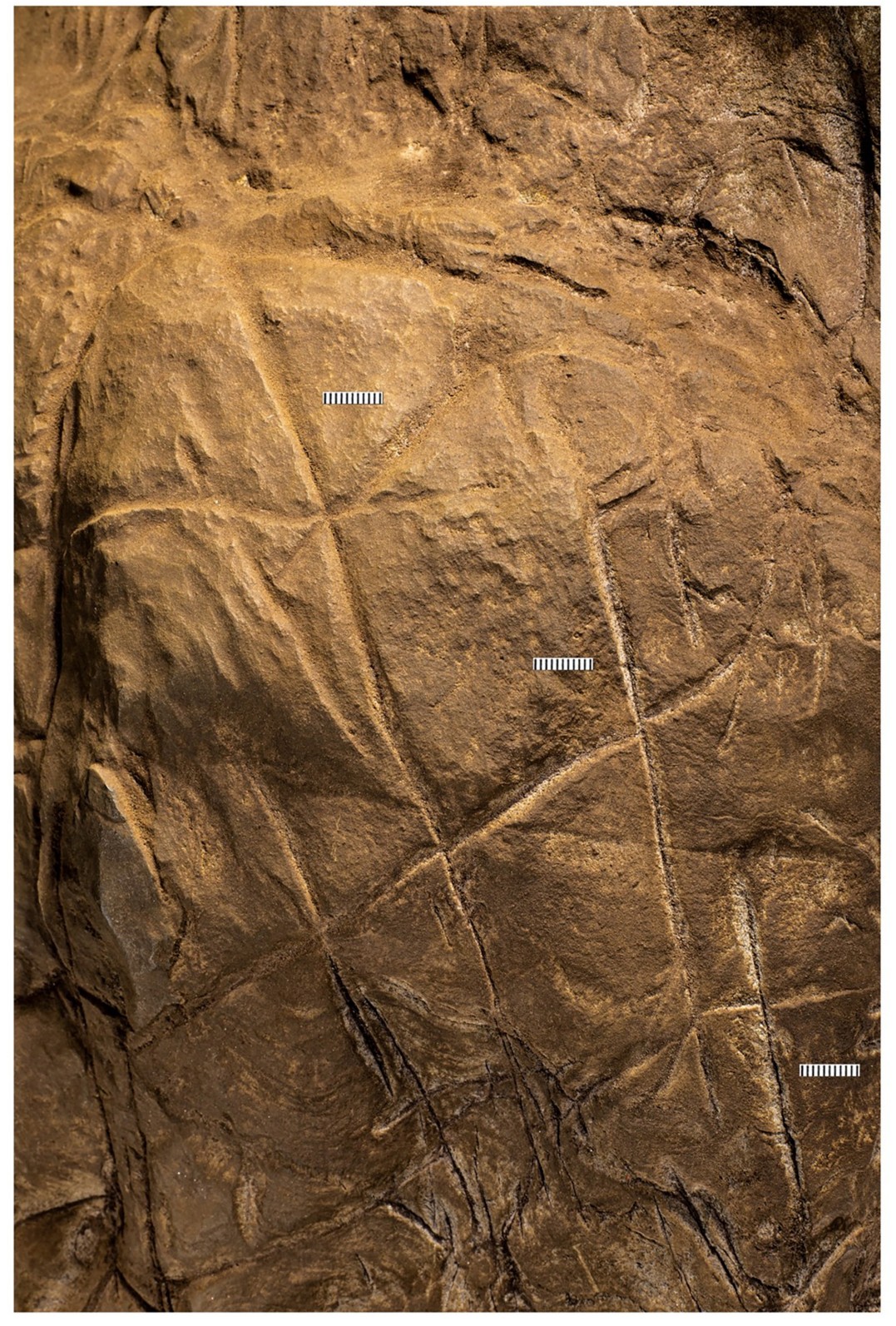

**Figure 12.** LED light images of the Panel A primary engravings. Scales in millimeters.

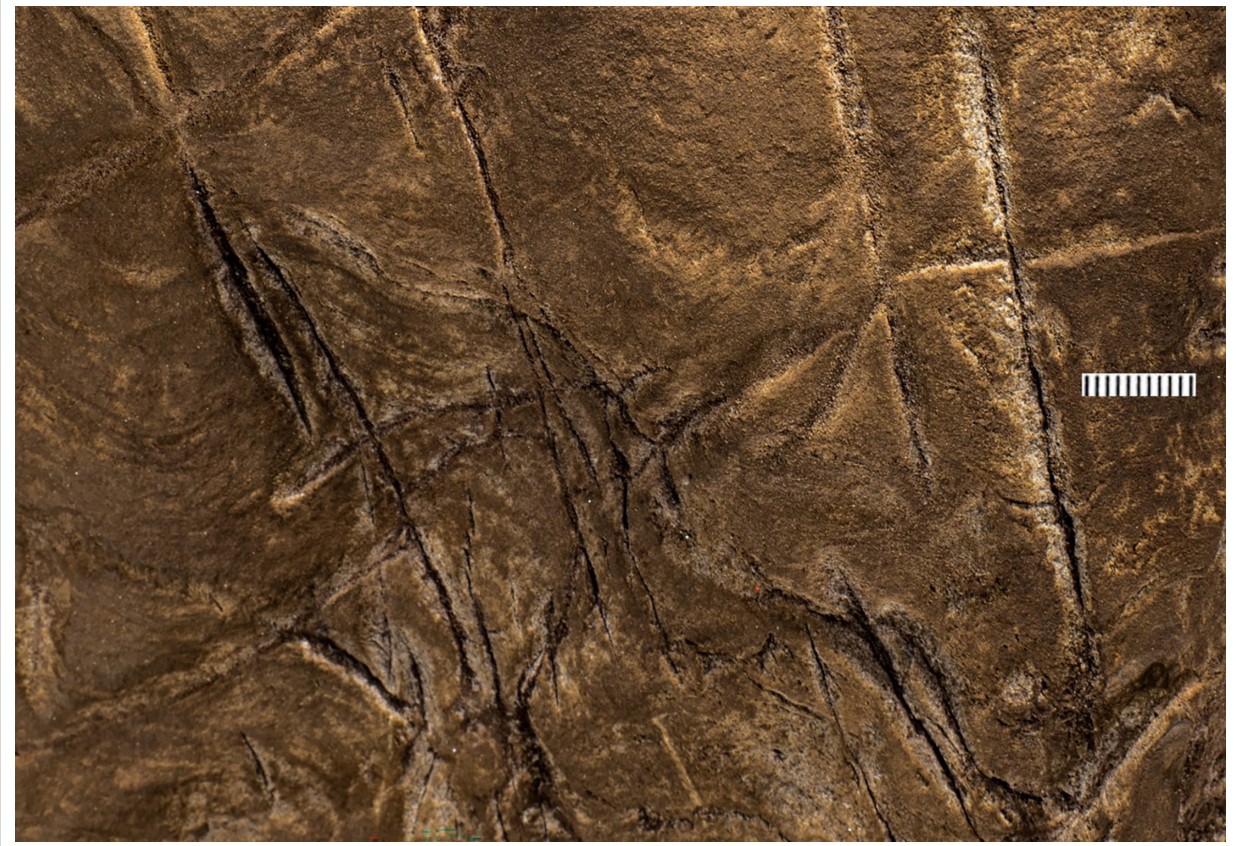

**Figure 13.** Lines 2, 6, 11, 14, 15, 16, 18, 21, 25, 27, 33, 38, 39, and 40. Note that etching 18 overcuts line 17 and line 30 overcuts line 25, indicating an order of creation. Note also that all lines engraved on the left side of the image cut through the fossil stromatolite visible as horizontal wavy lines in the rock.

in the form of geometric figures (*Figure 7*), crosses, X's and one possible non-linear geometric figure (*Figure 5*). It appears, in softer visible light, that a foreign substance has been applied to part of the panel. As was noted, the purpose of this article is not to describe these complex panels, but to simply note their presence in the Hill Antechamber. Future work in this difficult space is planned to sample the possible residues and map the non-natural etchings, attempt to date the etchings, and we will conduct experimental work on native dolomite in controlled experiments.

## Discussion

The context of these markings within the remote Dinaledi Subsystem, far from cave entrances and challenging to access (*Dirks et al., 2015*; *Kruger et al., 2016*; *Elliott et al., 2021*), suggests that the most parsimonious explanation for their manufacture is that *H. naledi* produced them (*Berger et al., 2015*; *Berger et al., 2025*). This is the only species that left any evidence of presence in this subsystem until the entry of modern cave explorers. In this discussion, we consider possible objections to this hypothesis and ways in which this hypothesis may be tested in the future. We also suggest some comparisons with the broader record of ancient markings beyond the Rising Star examples.

One possible objection is that some markings were not produced by hominins. For example, some have suggested that some of the apparent markings are consistent with weathered dolomite, while others may have been produced by non-hominin agents such as carnivores (*Martinón-Torres et al., 2024*). Linear features and other modifications of rock surfaces are sometimes produced by animal claws due to tunneling, climbing, or claw-sharpening, or other behaviors (*Bednarik, 1991*). Such features have often been observed within caves formed within relatively soft limestone formations, including those in some parts of Europe and southern Australia. Deeply incised claw markings are

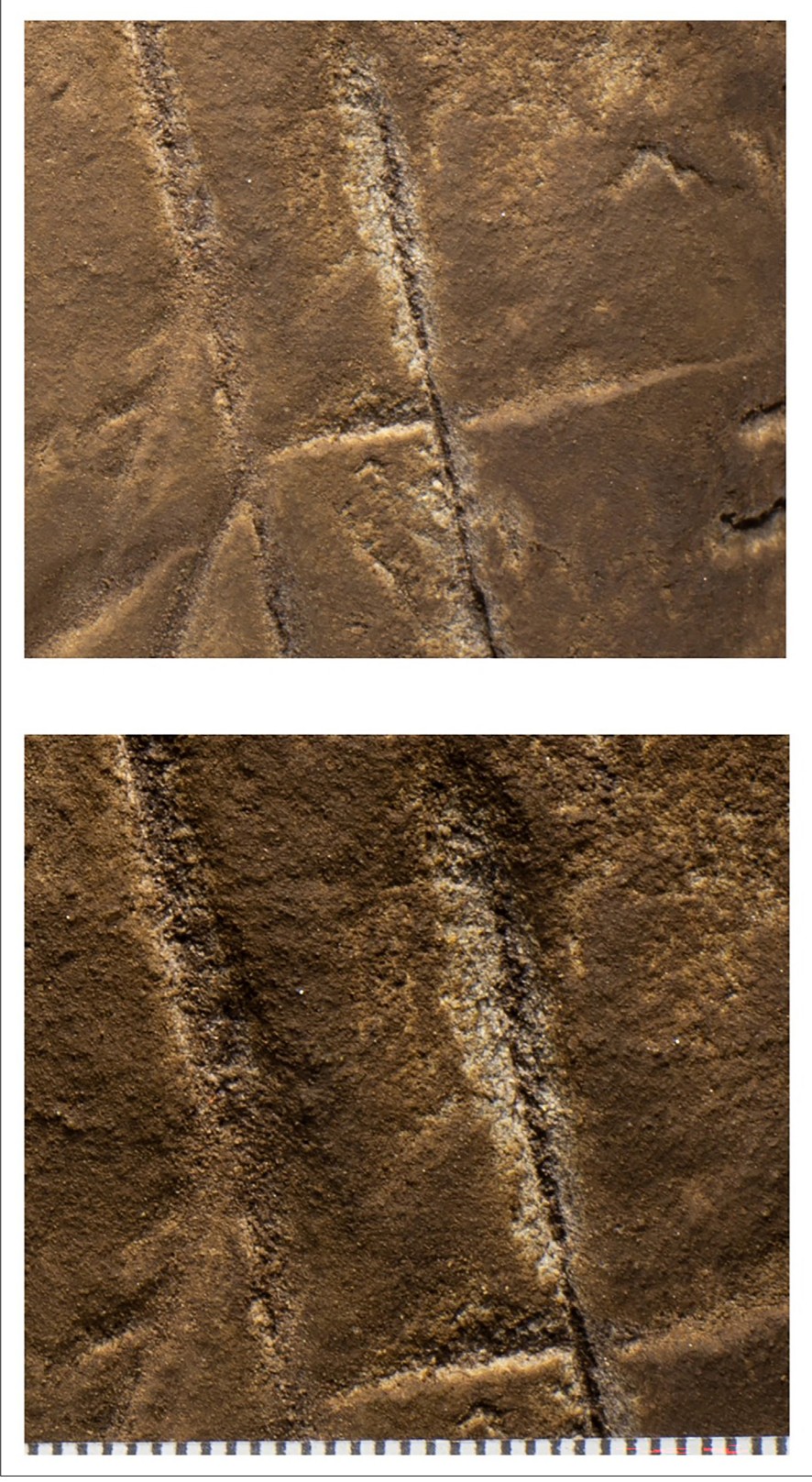

**Figure 14.** Magnified views of lines 6, 17, and 18. The bottom image is a slightly higher magnification of the top image. It is clear from these images that the engraving of line 17 preceded the engraving of line 18. Scale at bottom in millimeters.

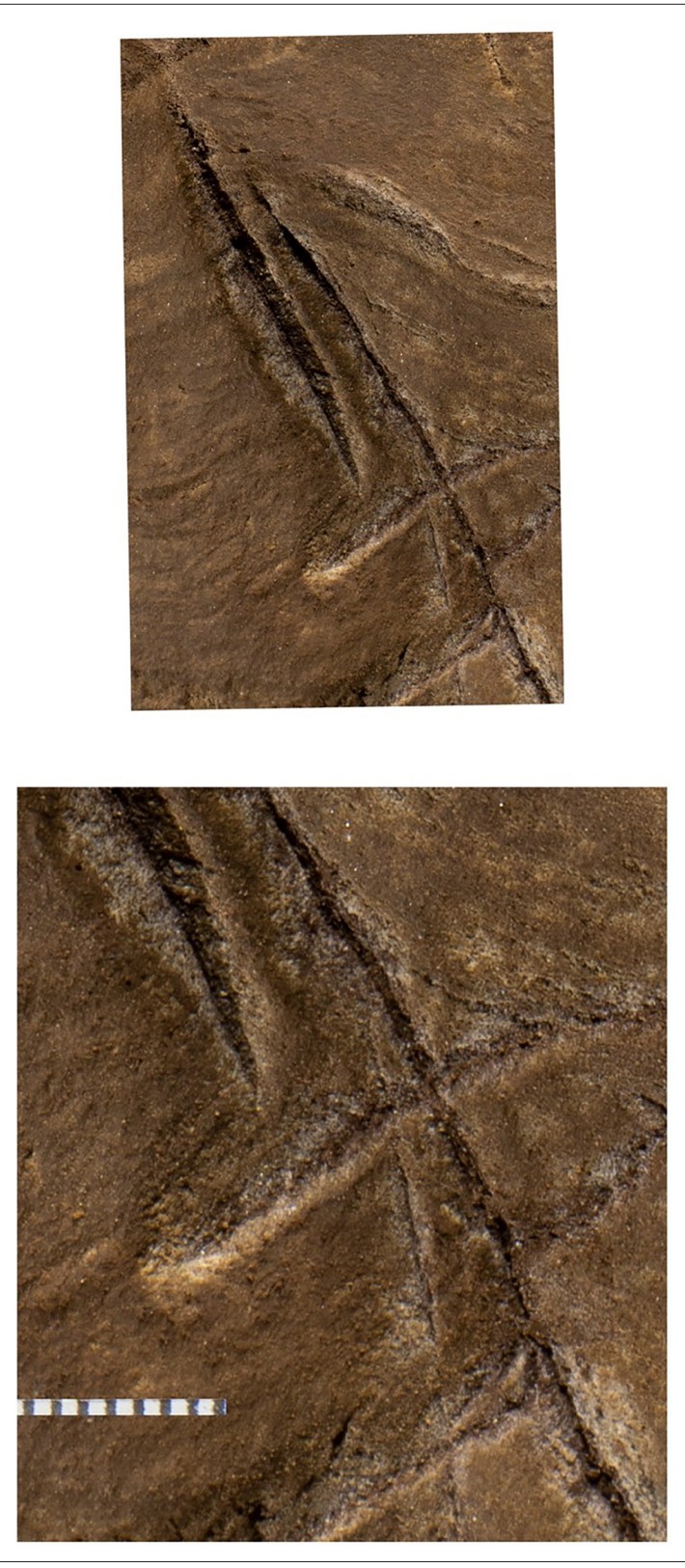

**Figure 15.** Magnified views of lines 16, 25, and 30. Note it is clear line 25 was etched first, followed by line 30. The lateral edge of line 31 can be seen at bottom left and the left edge of line 15 at top right. Note also the deep incision through the stromatolite layers by all lines. Scale in millimeters.

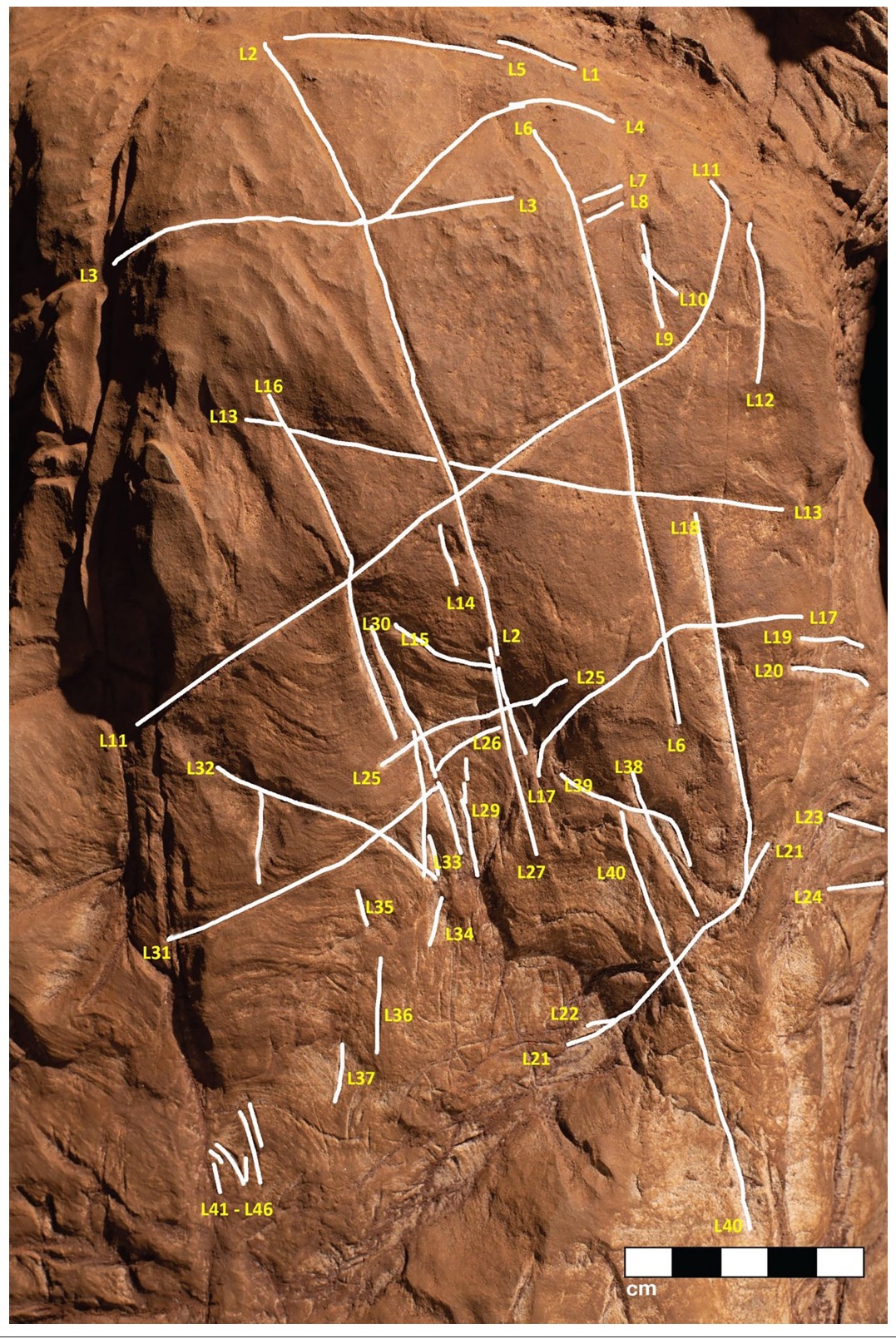

**Figure 16.** A conservative map of non-natural engravings observed on Panel A. Non-natural engravings are traced in white lines and given yellow number references in the text.

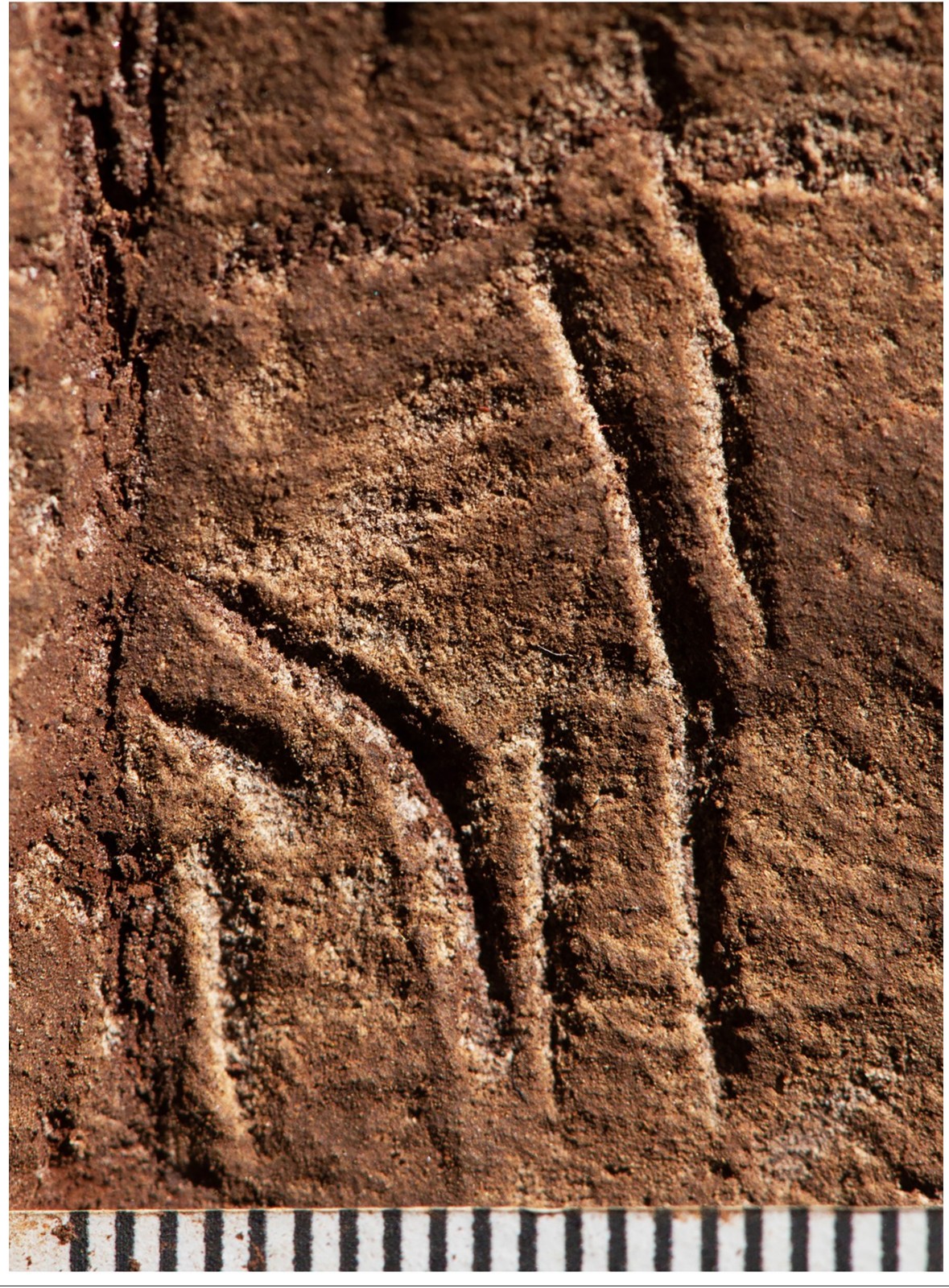

**Figure 17.** Magnified images of etchings 41 through 46 numbered from left to right. Scale in millimeters.

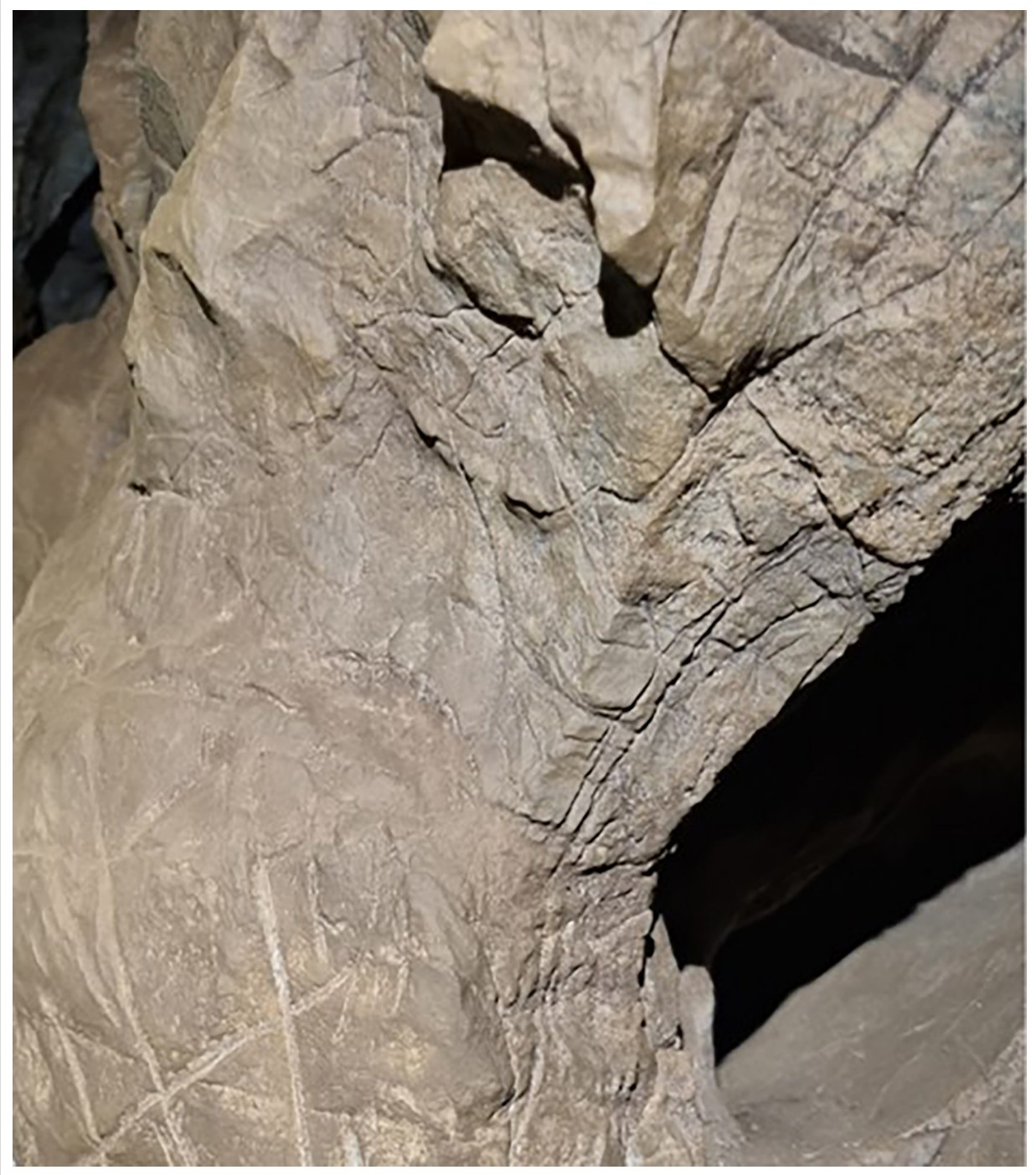

**Figure 18.** Image of dolomite above and right of Panel A. The top of the cross-hatched etchings can be seen in the lower left of the image. Note the smoothing and alteration of the panel's surface compared to natural, non-altered dolomitic surfaces above and right of the panel typical of unaltered surfaces throughout the system.

sometimes seen, which tend to include multiple parallel incisions with a spacing typical of the anatomical spacing between claws. When claw marks are made repeatedly on the same surface, they may cross each other or exhibit complex geometries. However, the susceptibility of a surface to marking by animal claws is determined by the hardness of the material. Animal claws have a hardness of 2–2.5 on Mohs hardness scale (*Rothschild et al., 2013*). Dolomite has a hardness of 3.5–4.5, with Malmani dolomites at the higher end of this scale, much harder than non-dolomitic limestones with hardness

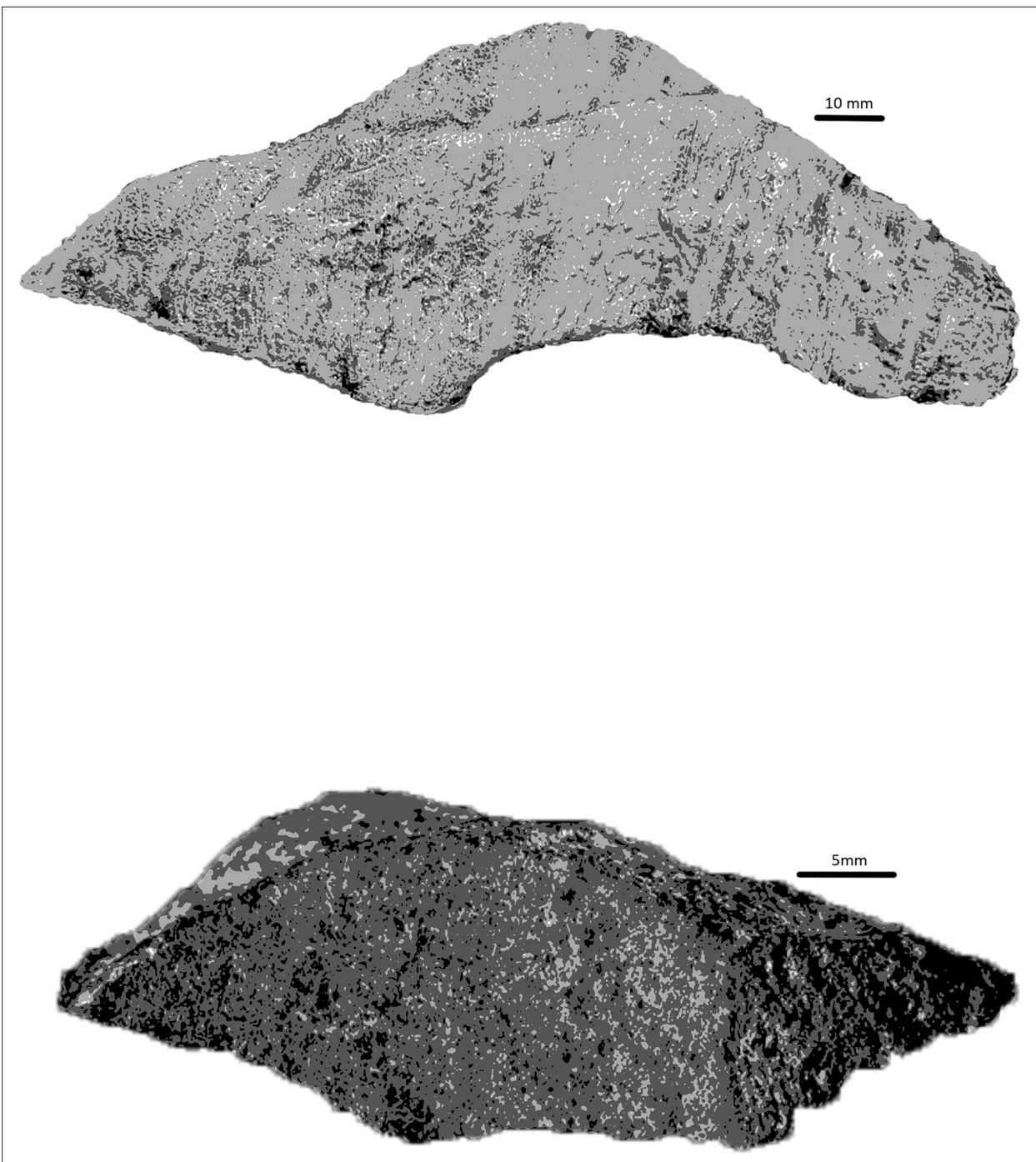

**Figure 19.** The tool-shaped artifact described in *Berger et al., 2025* (top) recovered from the Hill Antechamber burial immediately below Panels B and C compared to the artifact from Blombos Cave, South Africa, attributed by *Henshilwood, 2009* as having symbolic markings in ochre made by *Homo sapiens* circa 78k years ago.

of 2–3, which are commonly marked by animal claws. None of the cave systems in the Malmani dolomite have been noted for incised claw markings into dolomite itself of the depth and form seen in the Dinaledi marks (*Figure 17*), even though clawed species like porcupines, honey badgers, and leopards do use caves and do dig into softer sediments found in the area. The hardness of the rock does not admit scratches by animal claws of this kind; additionally, the specific features of Panel A do not exhibit the kind of spacing or morphology that would be expected for repeated scratching by animal claws, and their position well above the cave floor would not lend itself naturally to scratching

**Table 1.** Known humans who have entered the Dinaledi System (In approximate order of entry).

| | | | |
|---|---|---|---|
| 1 | Neil Ringdahl | 27 | Eric Roberts |
| 2 | Rick Hunter | 28 | Maropeng Ramalepa |
| 3 | Steven Tucker | 29 | Elliott Ross |
| 4 | John Dickie | 30 | Tebogo Makhubela |
| 5 | Selena Dickie | 31 | Mathabela Tsikoane |
| 6 | Bruce Dickie | 32 | Riaan Hugo |
| 7 | Matthew Dickie | 33 | Corey Jaskolski |
| 8 | Matthew Berger | 34 | Kenny Broad |
| 9 | Megan Berger | 35 | Juan Luis Arsuaga |
| 10 | Marina Elliott | 36 | Ignacio Martínez Mendizábal |
| 11 | Becca Peixotto | 37 | Carlos Lorenzo Merino |
| 12 | Lindsay Eaves Hunter | 38 | Rolf Quam |
| 13 | Hannah Morris | 39 | Keneiloe Molopyane |
| 14 | Elen Feuerriegel | 40 | Kerryn Warren |
| 15 | Alia Gurtov | 41 | Angharad Brewer-Gillham |
| 16 | Christo Saayman | 42 | Raymond Messitar-Tooze |
| 17 | Pieter Theron | 43 | Zubair Jinnah |
| 18 | Andre Doussy | 44 | Samuel Nkwe |
| 19 | Allen Herweg | 45 | Warren Smart |
| 20 | Michael Herweg | 46 | Lee Berger |
| 21 | Rupert Stander | 47 | Chris Collingridge |
| 22 | Lindin Mazilis | 48 | Ginika Ramsawak |
| 23 | Dirk van Rooyen | 49 | Sarah Johnson |
| 24 | Ashley Kruger | 50 | Agustín Fuentes |
| 25 | Zoë Rosen | 51 | Hipolito Collado Giraldo |
| 26 | Garrreth Bird | 52 | Sara Garcês |

in that location. Further, there is no evidence of carnivore presence within the subsystem: no carnivore remains, no evidence of carnivore burrowing activity, no coprolites, and no bones with tooth marks attributable to carnivores (*Dirks et al., 2015*; *Dirks et al., 2017*).

Another possible objection is that the engravings may be of comparatively recent origin, potentially made by modern cavers or other *H. sapiens* groups in Holocene or Late Pleistocene times (*Herries, 2023*; *Petraglia et al., 2023*; *Martinón-Torres et al., 2024*; *Pettitt and Wood, 2024*). This idea has two parts: one concerns the idea that other species or populations besides *H. naledi* were active within this subsystem; the other concerns the chronological placement of the markings relative to *H. naledi*. There is no evidence of any hominin activity in the Dinaledi Subsystem other than *H. naledi*. No remains attributable to other prehistoric hominin groups occur in this subsystem or any adjacent areas of the cave system, nor is there any other evidence of activity of such groups (*Dirks et al., 2015*; *Dirks et al., 2017*). Human cave explorers entered these areas only within the last 40 years, and the number of modern cavers and archaeologists who have entered the Dinaledi Subsystem is extremely limited (*Table 1*). There is no evidence of modern cavers altering cave walls in such a manner in the Dinaledi Subsystem, or elsewhere in the Rising Star system, and we have detected no evidence of traces of metallic tools in association with the markings. By contrast, abundant remains of *H. naledi* have been identified within the subsystem, including the Hill Antechamber Feature only 3–4 m away (*Berger*

*et al., 2025*). Nevertheless, the nature and composition of the tools that made these engravings are an area that can be tested with future work.

It is very challenging to establish the chronological age of engraved markings on rock surfaces (*Van, 1968*; *Butzer et al., 1979*; *Bednarik, 2002*; *Bednarik, 2008*; *Whitley, 2012*). Successful use of geochronological methods may occur when some or all of the markings were buried within sediment that can be dated (*Rodríguez-Vidal et al., 2014*), or when markings were later covered in part by mineral formations that can be dated using radiometric methods (*Hoffmann et al., 2016*), or when exposure to weathering creates a varnish that can be dated (*Dorn and Nobbs, 1992*; *Dragovich, 2000*; *Masson, 2006*; *Pillans and Fifield, 2013*). Still, even where geochronological methods are possible, they may provide only a minimum age, which may be thousands of years after the marks were produced. The markings that we have identified within the Dinaledi Subsystem are not overlain by naturally forming sedimentary deposition. Without further excavation of sediments that adjoin cave walls, we cannot say whether additional markings may exist in subsurface contexts. No obvious mineral deposition overlaps the engraved features. Our observations do note that there may be materials added to the surface of the engraved areas, and these may provide an avenue of investigation. The situation as a whole makes it challenging to build alternative lines of evidence about the age of the engravings beyond their spatial proximity to the *H. naledi* evidence from only a few meters away (*Berger et al., 2025*).

Thus, at present, the *H. naledi* skeletal material provides the only information about the chronology of hominin activity in this area of the cave system. The maximum age constraint reported by *Dirks et al., 2017* on *H. naledi* skeletal material (335 kyr BP) in Dinaledi is the highest 95% confidence limit of a direct ESR-US date on *H. naledi* teeth; while the minimum age constraint (241 kyr BP) is based on U-Th on a flowstone that formed in part around a bone fragment (*Dirks et al., 2017*; *Robbins et al., 2021*). These dates provide strong constraints on some *H. naledi* skeletal material from the Dinaledi Subsystem. Geochronological methods do not yet provide similar constraints for *H. naledi* material from other parts of the cave system, including the Lesedi Chamber (*Hawks et al., 2017*). Therefore, we cannot be certain that the constraints of the Dinaledi Subsystem evidence mark the full duration of *H. naledi* cultural activity across the cave system. However, we adopt as a conservative hypothesis that the age of these markings is constrained in the same way as the skeletal remains from the Dinaledi Subsystem, between 335 ka and 241 ka.

Stylistic criteria are also sometimes used to establish the chronology of cave art, although such comparisons should be accepted cautiously. In this case, the images are not typical of contemporary graffiti made by recent people. They are also not typical of rock art of the last several thousand years in southern Africa. The markings are comparable to other hominin-produced markings (*Figure 8* and *Figure 9*), pigment use, and curation of nonutilitarian objects from the later Middle Pleistocene of southern Africa. Small slabs of ironstone with incised lines were excavated from Wonderwerk Cave, with a minimum age of 276 kyr BP (*Beaumont and Vogel, 2006*; *Beaumont and Bednarik, 2013*; *Chazan and Horwitz, 2009*; *Jacobson et al., 2013*). One of these slabs bears a series of seven roughly parallel engraved lines, while crossed lines occur on other slabs (*Beaumont and Bednarik, 2013*; *Chazan and Horwitz, 2009*). *Beaumont and Bednarik, 2015* proposed that two sites in the southeastern Kalahari with open air rock engravings, Potholes Hoek and Nchwaneng, have highly eroded cupules dating to the early Middle Stone Age, possibly as early as 400 kyr. Small pebbles of quartz, chalcedony, and other materials occur within the Wonderwerk sequence and have been interpreted as manuports (*Beaumont and Vogel, 2006*; *Beaumont and Bednarik, 2013*), although an alternative hypothesis that they were ostrich gastroliths, potentially introduced by hominins who hunted the birds, may also be possible (*Tryon, 2010*). Mineral pigments including specularite and hematite have been found in a number of contexts dating to earlier than 300 kyr, including Kathu Pan 1, Wonderwerk, and Canteen Kopje (*Watts et al., 2016*). None of these examples of hominin-produced markings, pigment use, or other activities have been found near hominin fossil remains. Some of them predate any evidence of modern humans or early *Homo sapiens* anywhere on the continent.

The Dinaledi markings are comparable to some of the incised lines and engravings found at other sites in the later Middle and Late Pleistocene (*Figure 8* and *Figure 9*). Panel A gives the impression of overlapping crosses and lines that are remarkably similar in appearance to the engraving from Gorham's Cave, Gibraltar (*Rodríguez-Vidal et al., 2014*). This engraving was dated to greater than 39 kyr cal

BP and has been attributed to Neandertals. In addition to the incised slabs and cupules mentioned above, the later Pleistocene record of southern Africa includes engraved bones and engraved ochre blocks from Blombos Cave (d'*D'Errico et al., 2001*; *Henshilwood et al., 2002*; *Henshilwood et al., 2018*) and lines impressed within sand features that were later lithified into aeolianites (*Helm et al., 2021*). There are also a few other engravings from sites in Europe at similar time depth (*Von Petzinger, 2017*; *Kissel and Fuentes, 2017*, *Kissel and Fuentes, 2018a*), as well as geometric lines on a fresh-water mussel shell from Trinil, Java, attributed to *H. erectus* based on geochronological considerations (*Joordens et al., 2015*). The markings in the Dinaledi Subsystem share similarities with many of these geometric expressions from other sites and geographic regions. Several geometric forms identified by *Von Petzinger, 2017* are present, including cross-hatch, cruciform, line, flabelliform, scalariform, open angle, and oval. However, further analytic and comparative work must be conducted to confirm exactly how much similarity and overlap there is between the proposed Dinaledi engravings and the engravings at other Pleistocene sites where such designs have been found.

The precise location of the engraved markings within the Dinaledi Subsystem may be important to their interpretation. Many artifacts and rock walls from later Pleistocene sites with engraved marks appear to have a nonrandom placement of markings on an object or surface. *Henshilwood and Dubreuil, 2011* have suggested that one should be less focused on the specifics of the designs and rather concentrate on the underlying cause of their creation. Those and other authors have suggested symbolic implications for such engravings, under the assumption that they were made by *H. sapiens*. The broader record of engravings and other forms of meaning-making behavior in a range of hominin species and populations over the latter portions of the Pleistocene (*Kissel and Fuentes, 2018b*; *Dusseldorp and Lombard, 2021*; *Kissel and Fuentes, 2021*; *Fuentes et al., 2025*) suggests that such activity, be it 'symbolic' or not, was not exclusive to *H. sapiens*. The engravings reported here add to this growing dataset.

## Methods

As the main purpose of this work was documentation of the presence of engravings in proximity to the burials, for this study, the etchings and engraving markings were only examined using high-resolution photography and magnification of lines and markings (*Walderhaug Saetersdal, 2000*). Polarizing filters were also used to enhance relief, and this is indicated when used.

Cross-polarization was employed for control of specular highlights/reflections in order to limit artifacts when generating the 3D-depth map for photogrammetry purposes. A circular polarizer was used on the camera lens in conjunction with a linear polarizing gel placed over the two speed lights (electronic flash heads) used as the light source. The different minerals/material on the dolomite are reflecting/absorbing the cross-polarized light emphasizing the 'bright' striations visible in images.

Images were shot with a 50 mm (Ppolarizer fitted) at f/11 unless otherwise stated.

The light source used (twin speed lights with polarized gel attached) was placed as close to the lens axis as possible so that the angles of incidence approximate the reflected angles limiting shadow. This assisted us in building the 3D mesh for photogrammetry purposes. The cross-polarization also removed specular highlights that create artifacts.

We used Metashape 1.8.1 (Agisoft, Inc) to generate three-dimensional models of Panels A and B based on photographs taken with the parameters reported above. Generation of cross-sections and measurements from these models was performed with MeshLab 2021.20. Resolution of the three-dimensional surface is estimated to be accurate to 0.2 mm.

## Acknowledgements

Permits to conduct research in the Rising Star cave system were provided by the South African National Research Foundation (LRB). Permission to work in the Rising Star cave was given by the LRB Foundation for Research and Exploration. We acknowledge the funders of the various expeditions and documentation of the engravings, including the National Geographic Society (LRB), the Lyda Hill Foundation (LRB) and the National Research Foundation of South Africa (LRB). Laboratory work and travel were funded by the National Geographic Society (LRB), the Lyda Hill Foundation (LRB), the Fulbright Scholar Program (JH), the University of Wisconsin (JH), and Princeton University (AF).

# Additional information

## Funding

| Funder | Grant reference number | Author |
| --- | --- | --- |
| National Geographic Society | The Rising Star Project | Lee R Berger |
| John Templeton Foundation | The Rising Star Project | Lee R Berger<br>John Hawks |
| Lyda Hill Foundation | | Lee R Berger |
| Fulbright Scholar Program | | John Hawks |
| The University of Wisconsin | | John Hawks |
| Princeton University | | Agustín Fuentes |

The funders had no role in study design, data collection and interpretation, or the decision to submit the work for publication.

## Author contributions

Lee R Berger, Conceptualization, Resources, Funding acquisition, Investigation, Visualization, Methodology, Writing – original draft, Project administration, Writing – review and editing; John Hawks, Conceptualization, Investigation, Visualization, Methodology, Writing – original draft, Project administration, Writing – review and editing; Agustín Fuentes, Conceptualization, Investigation, Writing – original draft, Writing – review and editing; Dirk Van Rooyen, Supervision, Investigation, Writing – review and editing; Mathabela Tsikoane, Investigation, Visualization, Writing – review and editing; Maropeng Mpete, Samuel Nkwe, Investigation, Writing – review and editing; Keneiloe Molopyane, Conceptualization, Investigation, Methodology, Writing – original draft, Writing – review and editing

## Author ORCIDs

Lee R Berger ⓘ https://orcid.org/0000-0002-0367-7629
John Hawks ⓘ https://orcid.org/0000-0003-3187-3755
Agustín Fuentes ⓘ https://orcid.org/0000-0003-0955-8214

Reviewer #3 (Public review): https://doi.org/10.7554/eLife.89102.3.sa1
Reviewer #4 (Public review): https://doi.org/10.7554/eLife.89102.3.sa2
Author response https://doi.org/10.7554/eLife.89102.3.sa3

## Data availability

All data are included in the paper.

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
