## [Editor Report · eLife Assessment]

This article presents **important** information about potential *Homo naledi*-associated markings discovered on the walls of the Hill Antechamber of the Rising Star Cave system, South Africa. If confirmed, the antiquity, intentionality, and authorship of the reported markings will have profound archaeological implications, as such behaviors are otherwise widely considered to be unique to our species, *Homo sapiens*. This report concerns preliminary findings, and as it stands the study is **incomplete**, with further work needed in the future to support the claims about the anthropogenic nature, age, and author of the engravings.

---

## [Referee Report · Reviewer #3 (Public review)]

In a characteristically bold fashion, Lee Berger and colleagues argue here that markings they have found in a dark isolated space in the Rising Star Cave system are likely over a quarter of a million years old and were made intentionally by Homo naledi, whose remains nearby they have previously reported. As in a European and much later case they reference ('Neanderthal engraved 'art' from the Pyrenees'), the entangled issues of demonstrable intentionality, persuasive age and likely authorship will generate much debate among the academic community of rock art specialists. The title of the paper and the reference to 'intentional designs', however, leave no room for doubt as to where the authors stand, despite an avoidance of the word art, entering a very disputed terrain. Iain Davidson's (2020) 'Marks, pictures and art: their contributions to revolutions in communication', also referenced here, forms a useful and clearly articulated evolutionary framework for this debate. The key questions are: 'are the markings artefactual or natural?', 'how old are they?' and 'who made them?, questions often intertwined and here, as in the Pyrenees, completely inseparable. I do not think that these questions are definitively answered in this paper and I guess from the language used by the authors (may, might, seem etc) that they do not think so either.

Before considering the specific arguments of the authors to justify the claims of the title, we should recognise the shift in the academic climate of those concerned with 'ancient markings' that has taken place over the past two or three decades. Before those changes, most specialists would probably have expected all early intentional markings to have been made by *Homo sapiens* after the African diaspora as part of the explosion of innovative behaviours thought to characterise the 'origins of modern humans'. Now, claims for earlier manifestations of such innovations from a wider geographic range are more favourably received, albeit often fiercely challenged as the case for Pyrenean Neanderthal 'art' shows (White et al. 2020). This change in intellectual thinking does not, however, alter the strict requirements for a successful assertion of earlier intentionality by non-sapiens species. We should also note that stone, despite its ubiquity in early human evolutionary contexts, is a recalcitrant material not easily directly dated whether in the form of walling, artefact manufacture or potentially meaningful markings. The stakes are high but the demands no less so.

Why are the markings not natural? Berger and co-authors seem to find support for the artefactual nature of the markings in their location along a passage connecting chambers in the underground Rising Star Cave system. The presumption is that the hominins passed by the marked panel frequently. I recognise the thinking but the argument is weak. More confidently they note that "In previous work researchers have noted the limited depth of artificial lines, their manufacture from multiple parallel striations, and their association into clear arrangement or pattern as evidence of hominin manufacture (Fernandez-Jalvo et al. 2014)". The markings in the Rising Star Cave are said to be shallow, made by repeated grooving with a pointed stone tool that has left striations within the grooves, and to form designs that are "geometric expressions" including crosshatching and cruciform shapes. "Composition and ordering" are said to be detectable in the set of grooved markings. Readers of this and their texts will no doubt have various opinions about these matters, mostly related to rather poorly defined or quantified terminology. I reserve judgement, but would draw little comfort from the similarities among equally unconvincing examples of early, especially very early, 'designs'. Two or even three half convincing arguments do not add up to one convincing one.

The authors draw our attention to one very interesting issue: given the extensive grooving into the dolomite bedrock by sharp stone objects, where are these objects? Only one potential 'lithic artefact' is reported, a "tool-shaped rock [that] does resemble tools from other contexts of more recent age in southern Africa, such as a silcrete tool with abstract ochre designs on it that was recovered from Blombos Cave (Henshilwood et al. 2018)", also figured by Berger and colleagues. A number of problems derive from this comparison. First, 'tool-shaped rock' is surely a meaningless term: in a modern toolshed 'tool-shaped' would surely need to be refined into 'saw-shaped', 'hammer-shaped' or 'chisel-shaped' to convey meaning? The authors here seem to mean that the Rising Star Cave object is shaped like the Blombos painted stone fragment? But the latter is a painted fragment not a tool and so any formal similarity is surely superficial and offers no support to the 'tool-ness' of the Rising Star Cave object. Does this mean that Homo naledi took (several?) pointed stone tools down the dark passsageways, used them extensively and, whether worn out or still usable, took them all out again when they left? Not impossible, of course. And the lighting?

The authors rightly note that the circumstance of the markings "makes it challenging to assess whether the engravings are contemporary with the Homo naledi burial evidence from only a few metres away" and more pertinently, whether the hominins did the markings. Despite this honest admission, they are prepared to hypothesise that the hominin marked, without, it seems, any convincing evidence. If archaeologists took juxtaposition to demonstrate authorship, there would be any number of unlikely claims for the authorship of rock paintings or even stone tools. The idea that there were no entries into this Cave system between the Homo naledi individuals and the last two decades is an assertion not an observation and the relationship between hominins and designs no less so. In fact the only 'evidence' for the age of the markings is given by the age of the Homo naledi remains, as no attempt at the, admittedly very difficult, perhaps impossible, task of geochronological assessment, has been made.

The claims relating to artificiality, age and authorship made here seem entangled, premature and speculative. Whilst there is no evidence to refute them, there isn't convincing evidence to confirm them.

References:

Davidson, I. 2020. Marks, pictures and art: their contribution to revolutions in communication. Journal of Archaeological Method and Theory 27: 3 745-770.

Henshilwood, C.S. et al. 2018. An abstract drawing from the 73,000-year-old levels at Blombos Cave, South Africa. Nature 562: 115-118.

Rodriguez-Vidal, J. et al. 2014. A rock engraving made by Neanderthals in Gibralter. Proceedings of the National Academy of Sciences.

White, Randall et al. 2020. Still no archaeological evidence that Neanderthals created Iberian cave art.

Comments on latest version:

The authors have not modified their stance or the authority of their arguments since the original paper.

---

## [Referee Report · Reviewer #4 (Public review)]

Thank you for the opportunity to provide a peer-review of this manuscript, which I first reviewed in 2023 under the title of '241,000 to 335,000 Years Old Rock Engravings Made by Homo naledi in the Rising Star Cave system, South Africa'. My review is brief as the authors state they have made "relatively minimal changes", so most of the comments I made in 2023 still stand. Some of the language is a little more temperate but the main issues of this potentially landmark study remain and undermine scientific acceptance of the findings claim. The fact that this is an initial report does not excuse it from the normal conventions of building arguments supported by empirical data. Again, the absence of a rock art expert on the authorial team causes recurring weaknesses still to be evident (would one ask a rock art expert to analyse a new fossil hominin skull for example?). Specifically, there are two major issues that need to be resolved before there is necessary and sufficient cause to assign the term 'rock engravings' to the marks in the Dinaledi chamber. These are authorship and dating.

* Authorship: The assertion that the 'rock engravings' are anthropogenic remains unsupported by empirical evidence, with a number of possible natural factors that could just as likely have caused the marks. Not to use image enhancements - which is standard in most rock art research and has been for some time - is a critical omission. The concerns stated about AI and data standards are not developed and the authors are directed to the literature in this field, for example this 2025 overview - https://www.sciencedirect.com/science/article/pii/S1296207424002516. Again, having a rock art expert would show the AI concern to be valid but easily addressed using Data Standards. In the almost 2 years since the first pre-print was released, there has been ample time for high resolution photographs and scans of the purported 'rock engravings'; analysis of which by relevant experts could properly physically characterise the marks and thus establish more or less likely agents for their production. European-based researchers in particular has utilised this approach on material such as the Blombos ochre and marked bone from Europe and Africa. None of these methods is invasive or destructive.

To then go on and link Homo naledi to these markings is premature, especially when this landscape has been home to multiple hominins. Most rock art sites do not contain the physical bodily remains of their makers so we assign authorship based on dating (such as for Neanderthal era art in Europe for example); the second critical issue in this report:

* Dating: There is no direct or closely associated chronometric dating of the 'rock engravings' or their immediate context, so the age range claimed is unsupported. Rock art dating is notoriously difficult - and why researchers closely scrutinise dates produced. In this case, however, the chronological context is physically so far removed from these rock markings, as to be misleading at best and need to be discounted until a proper programme of dating has commenced. The sources cited for rock art dating tend to be out of date and it would be standard practice to have a geochronologist assess the rock-marked areas and then establish dating protocols.

Authorship and dating are cornerstone of archaeological/paleoanthropological work and need to established in the first instance. Until that has been done commensurate with current standards in global rock art research this potentially landmark finding cannot be taken as probable, only as possible. This is a pity as the last decade or so has revolutionised our understanding of the socially complex world multiple hominin species lived in, and marked in utilitarian and symbolic ways. The conditions for acceptance of ancient rock art has thus never been better, but the Dinaledi example needs to revisit research first principles around authorship and dating to be included as a credible part of this larger context. It would have been good to see a commitment to a coherent research programme to this end for this case study.

I hope these observations are useful. As above I keep them short as there has been minimal change to the 2023 ms, and my detailed comments on that remain with the first version of the work.

---

## [Author Response]

The following is the authors’ response to the original reviews:

We thank the reviewers for their very constructive and helpful comments on the previous version of this manuscript. They have focused on some important issues and have raised many valuable questions that we expect to answer as research begins on these markings. As has been often the case with preprints, a number of experts beyond the four reviewers and editor have provided comments, questions, and suggestions, and we have taken these on board in our revision of the manuscript. In particular, Martinón-Torres et al. (2024) focused several comments upon this manuscript and raise some points that were not considered by the reviewers, and so we discuss those points here in addition to the reviewer comments.

Some of us have been engaged in other aspects of the possible cultural activities of *Homo naledi*. After the discovery of these markings we considered it indefensible to publish further research on the activity of *H. naledi* within this part of the cave system without making readers aware that the *H. naledi* skeletal remains occur in a spatial context near markings on cave walls. Of course, the presence of markings leaves many questions open. A spatial context does not answer all questions about the temporal context. The situation of the Dinaledi Subsystem does entail some constraints that would not apply to markings within a more open cave or rock wall, and we discuss those in the text.

We find ourselves in agreement with most of the reviewers on many points. As reflected by several of the reviewers, and most pointedly in the remarks by reviewer 1, the purpose of this preprint is a preliminary report on the observation of the markings in a very distinctive location. This initial report is an essential step to enable further research to move forward. That research requires careful planning due to the difficulty of working within the Dinaledi Subsystem where the markings are located. This pattern of initial publication followed by more detailed study is common with observations of rock art and other markings identified in South Africa and elsewhere. We appreciate that the reviewers have understood the role of this initial study in that process of research.

Because of this, the revised manuscript represents relatively minimal changes, and all those at the advice of reviewers. Many thanks to all the reviewers for noting various typographic errors, missed references and other issues that we have done our best to fix in the revised manuscript.

Expertise of authors. Reviewer 4 mentions that the expertise of the authors does not include previous publication history on the identification of rock art, and other reviewers briefly comment that experts in this area would enhance the description. AF does have several publications on ancient engravings and other markings; LRB has geological training and field experience with rock art. Notwithstanding this, we do take on board the advice to include a wider array of subject experts in this research, and this is already underway.

Image enhancement. We appreciate the suggestions of some reviewers for possible strategies to use software filters to bring out details that may not be obvious even with our cross-polarization lighting and filtering. These are great ideas to try. In this manuscript we thought that going very far into software editing or image enhancement might be perceived by some readers as excessive manipulation, particularly in an age of AI. In future work we will experiment with the suggested approaches.

Natural weathering. In the process of review and commentary by experts and the public there has been broad acceptance that many of the markings illustrated in this paper are artificial and not a product of natural weathering of the dolomite rock. We deeply appreciate this. At the same time, we accept the comments from reviewers that some markings may be difficult to differentiate from natural weathering, and that some natural features that were elaborated or altered may be among the markings we recognize. On pages 3 and 4 we present a description of the process of natural subaerial weathering of dolomite, which we have rooted in several references as well as our own observations of the natural weathering visible on dolomite cave walls in the Rising Star cave system. This includes other cave walls within the Dinaledi Subsystem. We discuss the “elephant skin” patterning of natural dolomite surface weathering, how that patterning emerges, and how that differs from the markings that are the subject of this manuscript.

Animal claw marks. Martinón-Torres *et al.* 2024 accept that some of the markings illustrated on Panel A are artificial, but they offer the hypothesis that some of those markings may be consistent with claw marks from carnivores or other mammals. They provide a photo of claw marks within a limestone cave in Europe to illustrate this point. On pages 5 and 6 of the revised manuscript we discuss the hypothesis of claw marks. We discuss the presence of animals in southern Africa that may dig in caves or mark surfaces. However the key aspect of the Malmani dolomite caves is that the hardness of dolomitic limestone rock is much greater than many of the limestone caves in other regions such as Europe and Australia, where claw marks have been noted in rock walls. As we discuss, we have not been able to find evidence of claw marks within the dolomite host bedrock of caves in this region, although carnivores, porcupines, and other animals dig into the soft sediments within and around caves. The form of the markings themselves also counter-indicates the hypothesis that they are claw marks.

Recent manufacture. One comment that occurs within the reviews and from other readers of the preprint is that recent human visitors to the cave, either in historic or recent prehistoric times, may have made these marks. We discuss this hypothesis on page 6 of the revised manuscript. The simple answer is that no evidence suggests that any human groups were in the Dinaledi Subsystem between the presence of *H. naledi* and the entry of explorers within the last 25 years. The list of all explorers and scientific visitors to have entered this portion of the cave system is presented in a table. We can attest that these people did not make the marks. More generally, such marks have not been known to be made by cavers in other contexts within southern Africa.

Panels B and C. We have limited the text related to these areas, other than indicating that we have observed them. The analysis of these areas and quantification of artificial lines does not match what we have done for the Panel A area and we leave these for future work.

Presence of modern humans. We have observed no evidence of modern humans or other hominin populations within the Dinaledi Subsystem, other than *H. naledi*. Several reviewers raise the question of whether the absence of evidence is evidence of absence of modern humans in this area. This is connected by two of the reviewers to the observation that the investigation of other caves in recent years has shown that markings or paintings were sometimes made by different groups over tens of thousands of years, in some cases including both Neanderthals and modern humans. We have decided it is best for us not to attempt to prove a negative. It is simple enough to say that there is no evidence for modern humans in this area, while there is abundant evidence of *H. naledi* there.

Association with *H. naledi*. Reviewer 2 made an incisive point that the previous version contained some text that appeared contradictory: on the one hand we argued that modern humans were not present in the subsystem due to the absence of evidence of them, yet we accepted that *H. naledi* may have been present for a longer time than currently established by geochronological methods.

We appreciate this comment because it helped us to think through the way to describe the context and spatial association of these markings and the skeletal remains, and how it may relate to their timeline. Other reviewers also raised similar questions, whether the context by itself demonstrates an association with *H. naledi*. We have revised the text, in particular on pages 5 and 7, to simply state that we accept as the most parsimonious alternative at present the hypothesis that the engravings were made by *H. naledi,* which is the only hominin known to be present in this space.

Age of *H. naledi* in the system. At one place in the previous manuscript we indicated that we cannot establish that *H. naledi* was only active in the cave system within the constraints of the maximum and minimum ages for the Dinaledi Subsystem skeletal remains (viz., 335 ka – 241 ka), because some localities with skeletal material are undated. We have adjusted this paragraph on page 7 to be clear that we are discussing this only to acknowledge uncertainty about the full range of *H. naledi* use of the cave system.

Geochronological methods. Several reviewers discuss the issue of geochronology as applied to these markings. This is an area of future investigation for us after the publication of this initial report. As some reviewers note, the prospects for successful placement of these engraved features and other markings with geochronological methods depends on factors that we cannot predict without very high-resolution investigation of the surfaces. We have included greater discussion of the challenges of geochronological placement of engravings on page 6, including more references to previous work on this topic. We also briefly note the ethical problems that may arise as we go further with potentially invasive, destructive or contact studies of these engravings, which must be carefully considered by not just us, but the entire academy.

Title. Some reviewers suggested that the title should be rephrased because this paper does not use chronological methods to derive date constraints for the markings. We have rephrased the title to reflect less certainty while hopefully retaining the clear hypothesis discussed in the paper.